# BARCODE: high throughput screening and analysis of soft active materials

Qiaopeng Chen[1,5], Aditya Sriram [2,5], Ayan Das[1], Katarina Matic[2], Maya Hendija[2], Keegan Tonry[3], Jennifer L. Ross [4], Moumita Das [3], Ryan J. McGorty [2], Rae M. Robertson-Anderson [2,6] ✉ & Megan T. Valentine [1,6] ✉

Active, responsive, non-equilibrium materials–at the forefront of materials engineering–offer dynamical restructuring, mobility and other complex life-like properties. Yet, this enhanced functionality comes with significant amplification of the size and complexity of the datasets needed to characterize their properties, thereby challenging conventional approaches to analysis. To meet this need, we present BARCODE: Biomaterial Activity Readouts to Categorize, Optimize, Design and Engineer, an open-access software that automates high throughput screening of microscopy video data to enable non-equilibrium material optimization and discovery. BARCODE produces a unique fingerprint or 'barcode' of performance metrics that visually and quantitatively encodes dynamic material properties with minimal file size. Using three complementary material-agnostic analysis branches, BARCODE significantly reduces data dimensionality and size, while providing rich, multiparametric outputs and rapid tractable characterization of activity and structure. We analyze a series of datasets of cytoskeleton networks and cell monolayers to demonstrate BARCODE's abilities to accelerate and streamline screening and analysis, reveal unexpected correlations and emergence, and enable broad non-expert data access, comparison, and sharing.

Biological activity manifests through a wide range of mechanical and dynamical features, such as stiffening, restructuring and flow. Examples include migrating slime molds, developing tissues, and the cytoskeleton, a protein network in living cells that undergoes dramatic remodeling during cell division, crawling, and wound healing[1–7]. Engineered materials can recapitulate many such features, and offer stimuli-responsiveness, patterning, and spatiotemporal control that could, in principle, be harnessed for applications such as self-healing infrastructure, dynamic prosthetics, and self-sensing protective barriers[8–19]. Like their biological analogs, engineered materials exhibiting such functionalities must be multicomponent and multiphase,

with structures and dynamics operating over a broad range of length and time scales. For example, in vitro networks that recapitulate properties of the cell cytoskeleton, such as contractility, self-organization, and responsivity to external stimuli, comprise filamentous proteins, such as actin and microtubules, enzymatically-active motor proteins, such as myosin and kinesin, and a host of crosslinking proteins[12,20–30]. Varying the network formulation and intermolecular interactions can tune the material dynamics and structure over decades of spatiotemporal scales[20,25,29,31–33]. Similarly, cells in vitro exhibit dynamics and restructuring similar to living tissues, such as jamming, flow, and collective patterns emerging on length scales much longer

---

[1]Department of Mechanical Engineering, University of California, Santa Barbara, CA, USA. [2]Department of Physics and Biophysics, University of San Diego, San Diego, CA, USA. [3]School of Physics and Astronomy, Rochester Institute of Technology, Rochester, NY, USA. [4]Department of Physics, Syracuse University, Syracuse, NY, USA. [5]These authors contributed equally: Qiaopeng Chen, Aditya Sriram. [6]These authors jointly supervised this work: Rae M. Robertson-Anderson, Megan T. Valentine. ✉e-mail: randerson@sandiego.edu; valentine@engineering.ucsb.edu

than the individual agents driving the motion[34–39]. This dynamic structural heterogeneity, foundational to function, challenges classic approaches to material design, characterization and deployment.

Complex active dynamics and restructuring present several challenges to developing predictive relationships between material formulation and performance, and to realizing tractable engineering designs. Video sizes and complexity limit data sharing and use of standard software to process images within manageable time frames, often forcing valuable information to be ignored, discarded, or siloed. Active materials require a higher dimensionality of characterization metrics, which often emerge in unexpected ways and on spatio-temporal scales that are difficult to predict a priori. Inconsistencies in metrics, definitions, and approaches further hinder the identification of performance intersections among materials. These complexities demand readily accessible material-agnostic algorithms that enable rapid screening of large datasets and characterization of emergent dynamics in a manner that enables data- and physics-driven modeling and rational material design.

To address these challenges, we present BARCODE: Biomaterial Activity Readouts to Categorize, Optimize, Design and Engineer, an open-access software to facilitate the democratized discovery and optimization of non-equilibrium materials. BARCODE automates high throughput (HTP) screening of optical microscopy videos and produces a unique fingerprint that encodes dynamic material properties (Fig. 1). Consisting of three complementary branches that leverage standardized and widely-used image analysis approaches (Fig. 1D), BARCODE produces a unique array or 'barcode' of performance metrics for each video, and the collective dataset (Fig. 1E), significantly reducing data dimensionality, complexity and size, while providing rich, multiparametric outputs. Importantly, screening is performed without consideration of material composition or formulation, allowing unexpected correlations between performance metrics or disparate material systems to be rapidly revealed. To produce each barcode, the software also calculates rich reduced data structures (RDS) that enable more detailed understanding of the time-evolving material structures and mechanics, and archives the RDS to enable subsequent hypothesis-driven research. Through these features, BARCODE not only streamlines on-system screening and accelerates analysis but also enables non-expert data access and sharing across different materials and communities.

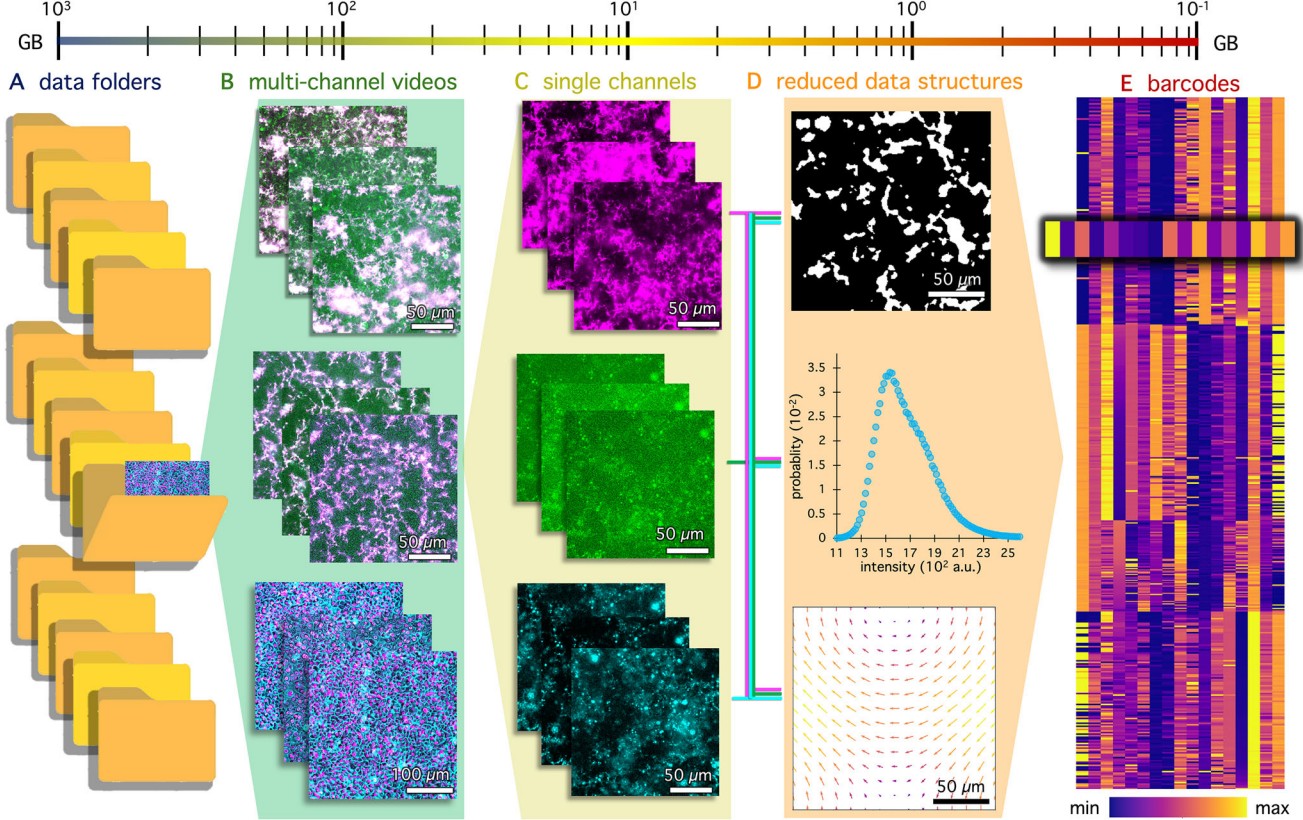

**Fig. 1 | BARCODE performs HTP screening of large video-based datasets to extract an information-rich, sparse-data barcode that reduces dataset size by 4 orders of magnitude. A** In typical experiments, a large number of multi-channel microscopy videos are acquired and saved in a hierarchical folder structure with (**B**) individual multi-channel videos within each folder, representing various compositions of active materials. Shown from top to bottom are kinesin-driven composites of actin (green) and microtubules (magenta) with (top) and without (middle) passive crosslinkers; and (bottom) a monolayer of cells with separately labeled nuclei (magenta) and cytoplasm (cyan). **C** Each multichannel video is separated into stacks of single-channel images. Shown from top to bottom are microtubules, actin, and microtubule crosslinkers (ASE1, anaphase spindle elongation protein 1) separated from the merged images shown in (**B**) (top). **D** Each channel is fed into three independent analysis branches that leverage vetted image analysis methods: binarization (top), pixel intensity distributions (middle), and optical flow (bottom). The reduced data structures (RDS) that each branch computes are saved for further screening and hypothesis-driven research outside of BARCODE; and used to extract a low dimensional 1×17 matrix or 'barcode' which visually and quantitatively represents key output metrics characterizing material structure, reconfiguration and dynamics. **E** Each video channel is reduced to a unique barcode that is compiled into a consolidated array that provides a comprehensive and quantitative fingerprint of the dataset. The different colors depict the values of each metric from the minimum (blue) to maximum (yellow) assigned values using the plasma colorscale, as shown by the color scale bar.

**Table 1 | BARCODE Parameters fully described in the main text and SI Section 1**

| Parameter | Description |
| --- | --- |
| Connectivity $C$ | Fraction of frames in a binarized video in which material is percolated across at least one dimension |
| Maximum Island Area $I$ | Fractional area of largest contiguous region of white pixels across all frames of a binarized video |
| Maximum Void Area $V$ | Fractional area of largest contiguous region of black pixels across all frames of a binarized video |
| Maximum Island Area Change $\Delta I$ | Relative change in maximum island area between the initial and final $X$% of frames of a binarized video |
| Maximum Void Area Change $\Delta V$ | Relative change in maximum void area between the initial and final $X$% of frames of a binarized video |
| Initial Maximum Island Area $I_{0,1}$ | Area of largest island in the initial $X$% of frames of a binarized video |
| Initial 2nd Maximum Island Area $I_{0,2}$ | Area of second largest island in the initial $X$% of frames of a binarized video |
| Maximum Kurtosis $K$ | Largest intensity distribution kurtosis value across all frames of a video |
| Maximum Median Skewness $S_1$ | Largest median intensity distribution skewness value across all frames of a video |
| Maximum Mode Skewness $S_2$ | Largest mode intensity distribution skewness value across all frames of a video |
| Kurtosis Change $\Delta K$ | Change in maximum kurtosis between the initial and final $X$% of frames of a video |
| Median Skewness Change $\Delta S_1$ | Change in maximum median skewness between the initial and final $X$% of frames of a video |
| Mode Skewness Change $\Delta S_2$ | Change in maximum mode skewness between the initial and final $X$% of frames of a video |
| Speed $v$ | Average magnitude of velocity vectors across all $p{\times}p$ windows of all $T/k$ flow fields of a video |
| Speed Change $\Delta v$ | Change in average speed between the initial and final $X$% of flow fields of a video |
| Mean Flow Direction $\theta$ | Average direction of velocity vectors across all $p{\times}p$ windows of all $T/k$ flow fields of a video |
| Directional Spread $\sigma_\theta$ | Circular standard deviation of velocity vector directions across all $p{\times}p$ windows of all $T/k$ flow fields of a video |

## Results

### BARCODE: Biomaterial Activity Readouts to Categorize, Optimize, Design and Engineer

BARCODE is designed to enable rapid HTP analysis of multi-channel microscopy videos to screen for desirable features of active material systems (Fig. 1). By extracting key parameters that describe material structure and dynamics (Table 1), BARCODE reduces large (~1 TB) and complex datasets (Fig. 1A-C) by up to four orders of magnitude. BARCODE's core architecture executes three complementary, yet independent, 'branches' in parallel (Fig. 1D). We designed each branch to leverage established image processing tools–image binarization (IB), pixel intensity distribution (ID) analysis, and optical flow (OF)[40]–for which there is an extensive body of literature describing approaches, implementation, and best practices[41-47]. Each branch produces distinct metrics selected to encode key structural and dynamic features of active materials, organized in a 1×17 'barcode' that is output both numerically and as a color-coded array (Fig. 1E). We compute all 17 metrics (Table 1) using standardized reduced data structures (RDS) (Fig. 1D)–binarized video images (IB), pixel intensity distributions (ID), and velocity fields (OF) – which are automatically produced and archived to facilitate future downstream analysis and that contain substantially more information than the 17-parameter barcode itself. We envision the barcodes will be used primarily for initial assessment and rapid down-selection, and the standardized outputs of the RDS will facilitate more time-intensive and hypothesis-driven downstream analyses. Based on our extensive analysis of active matter systems, we found that BARCODE's reduced set of 17 simple metrics captures key dynamic and structural information during rapid screens of large datasets. However, the software is material-agnostic, modular and highly adaptable: branches and metrics can be easily added or removed without impacting other metrics, thus providing flexibility, while allowing for discovery of unexpected correlations or trends.

As fully described in the Supplemental Information (SI, Section 1), BARCODE is a Python-based package that reads .tif and .nd2 video files and converts the data into arrays with dimensions $(T, m, n, c)$, where $T$ is the number of frames, $m$ and $n$ are the number of pixels along the horizontal and vertical axis of each frame, and $c$ is the number of channels (for, e.g., confocal videos with multiple components of a material labeled with distinct fluorophores and recorded in separate detectors). The software has a user-friendly graphical user interface (GUI) and several adjustable parameters that the user can set to tailor and optimize the operations of each branch for their system. We also provide a detailed online tutorial[48] to guide users in choosing parameters that best suit their data. BARCODE outputs include three RDS files, one for each branch, for every video, saved as .csv files; and a .csv file and colorized .svg with the 17 BARCODE metrics for the entire dataset (e.g., Figs. 1E, 2E). BARCODE can be executed on dozens, even hundreds of videos to produce a single barcode array (Fig. 1E), in 1–4 min per GB (SI Table S1), that can be used to identify patterns, correlations and trends.

### Demonstrating BARCODE workflow and utility

To demonstrate the BARCODE workflow, we analyze two representative videos of motor-driven actin-microtubule composites displaying distinct structures and dynamics (Fig. 2A).

The Image Binarization (IB) branch (Fig. 2B) converts each grayscale image of size $m \times n$ pixels of the video of $T$ frames into a binary image of white (1) and black (0) pixels. The threshold pixel intensity for binarization is computed from a user-specified offset that provides a threshold percentage $\%I$ above the mean intensity of each image (Fig. 2B, top). This process assigns a unique threshold intensity value for each frame, set by its mean value, which corrects for reduced mean intensities of images over time due to photobleaching and other spurious fluctuations in intensity. We assume in this analysis that there are minimal spatial variations in background intensity across the field of view. To increase processing speed and reduce the data size, the resulting stack of binarized images, which is a saved RDS, can be downsampled in time, by choosing to analyze every $k^{th}$ frame, and spatially, by averaging together pixels within $p \times p$ windows to result in a stack of $T/k$ images of $m/p \times n/p$ pixels. We have found that $\%I = 10$, $k = 10$ and $p = 2$-8 provide sufficient resolution and accuracy while maintaining rapid processing times (SI Table S1).

To characterize key structural features of the system, we calculate the areas of connected regions of white pixels, 'islands', and black pixels, 'voids' in each frame of the binarized RDS. The islands are regions of the field of view (FOV) where the material resides while voids are absent of material, and we compute their areas relative to the

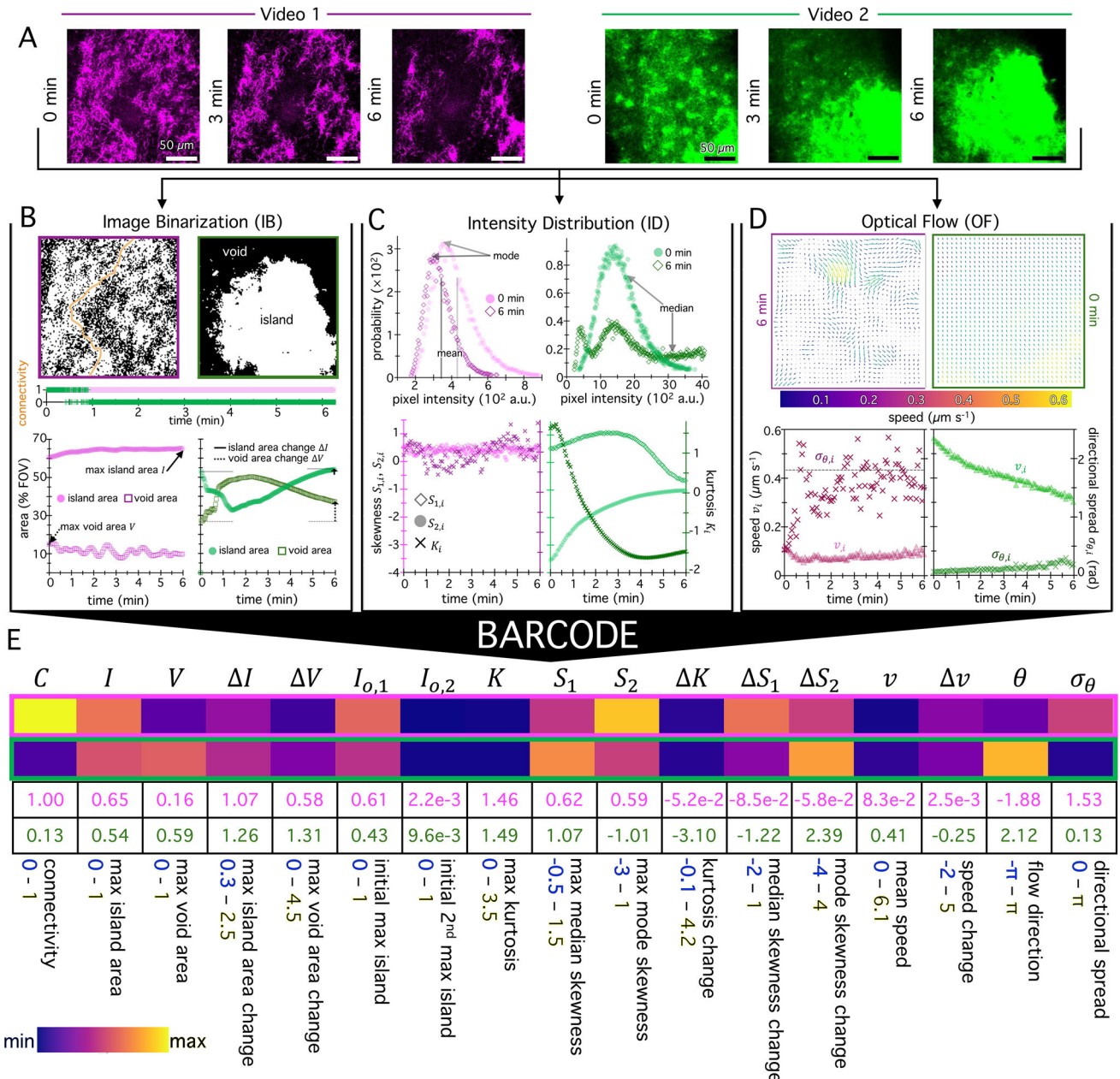

**Fig. 2 | BARCODE leverages three independent analysis branches to rapidly extract a structural and dynamical fingerprint. A** Select frames at beginning (0 min), middle (3 min) and end (6 min) of confocal fluorescence microscopy videos show microtubules (magenta, Video 1) and actin (green, Video 2). **B** The Image Binarization (IB) branch binarizes every $k^{th}$ frame, resulting in down-sampled binary image stacks. Binarized 0 min and 6 min frames of Video 1 (magenta border) and Video 2 (green border), show full connectivity (orange line), and largest island and void (contiguous white and black region). Connectivity $C_i$ vs time for Video 1 (magenta) and 2 (green) yields the fractional connectivity $C$. Maximum island and void areas, $I_i$ and $V_i$, versus time determine: maximum island and void areas, $I$ and $V$, and their changes between the initial and final 5% of frames, $\Delta I$ and $\Delta V$. **C** The Intensity Distribution (ID) branch evaluates the distribution of pixel intensities for every $k^{th}$ frame, displayed for the initial and final frames of Video 1 (magenta) and 2 (green) with mode, mean and median, indicated. Skewness $S_{2,i}$

(left axis, filled circles) and kurtosis $K_i$ (right axis, open diamonds) vs time for every $k^{th}$ frame yields maximum kurtosis $K$, maximum mode skewness $S_2$ and their changes between the initial to final 5% of frames, $\Delta K$, $\Delta S_2$. **D** The Optical Flow (OF) branch generates a velocity field for every $k^{th}$ frame, downsampled by a $p \times p$ pixel window (default $p = 8$). The last (6 min) and first (0 min) flow fields are shown for Video 1 (left, magenta border) and 2 (right, green border), respectively, with arrow length and color indicating speed relative to individual flow fields and the group (see colorscale). Mean speed $v_i$ and directional spread $\sigma_{\theta,i}$ evaluated for every $k^{th}$ frame yields speed $v$, speed change $\Delta v$ and directional spread $\sigma_\theta$. **E** A 1×17 'barcode' of metrics, each labeled by its variable and name (Table 1), for Video 1 (magenta border) and 2 (green border). Numeric values are given in magenta or green, respectively. Barcode colorscale is bounded by metric-specific minimum (blue outlined text) and maximum (yellow outlined text) values.

FOV area, with values ranging from 0 to 1. We identify the maximum island area $I_i$ and maximum void area $V_i$ in each frame $i$ (Fig. 2B bottom) and report in BARCODE the global maximum island ($I$) and void ($V$) area, indicators of the size of the largest features and gaps in the material. In general, to produce a more statistically robust measure of

any maximum barcode metric, we compute and report the mean of the highest 10% of all values over all frames. To characterize the initial structure of the time-evolving material, we also compute the average initial maximum and initial secondary maximum island areas in the first $X$% of frames of each video $I_{0,1}$ and $I_{0,2}$, where $X$ is a user-defined value

with default value $X = 5$. For materials that are primarily space-spanning or connected, $I_{0,1}$ will be much larger than $I_{0,2}$, while materials that comprise more homogenous distributions of independent entities (e.g., clusters, particles) will have $I_{0,1} \approx I_{0,2}$. Another important feature of a material that often dictates its behavior and mechanics is the degree to which it is percolated across dimensions, which we assess by computing 'connectivity'. The connectivity per frame $C_i$ is defined as $C_i = 1$ (or 0) for images having a continuous path (or not) of white pixels extending from edge to edge along at least one axis. In BARCODE, we report the fraction of frames that are connected $C = \langle C_i \rangle$. To assess any time-dependent restructuring of the material, we also provide in BARCODE the relative change in the island and void areas from the first $X\%$ of frames to the last $X\%$ ($\Delta I$, $\Delta V$), which we compute relative to the initial value, such that changes that are greater than (or less than) one correspond to increased (or decreased) area. We have found $X = 5$ to be optimal for capturing dynamic information while ensuring sufficient signal to noise. The 7 metrics of the IB branch (Fig. 2E) accurately describe the time-varying structures shown in Fig. 2A. Video 1 remains connected ($C = 1$) with a much smaller maximum void area as compared to maximum island area ($I > V$) and the void area shrinks over time ($\Delta V < 1$), indicative of restructuring to more space-filling composition. Conversely, Video 2 is poorly connected ($C = 0.13$), has comparable maximum island and void sizes ($I \approx V$) and shows a large increase in void area ($\Delta V > 1$) over the time course of observation.

The Intensity Distribution (ID) branch (Fig. 2C) evaluates the probability distribution of pixel intensities, which serves as a proxy for mass density, and outputs 6 BARCODE entries that describe the global distribution features, and how these features change from the beginning to the end of the video to report aspects of time-dependent restructuring. The ID branch produces distributions for all $T$ frames in a video, which we output as an RDS. Similar to the IB branch, to accelerate processing speed, users can also choose to only evaluate a subset of $T/k$ frames where the interval $k$ is specified by the user. We have found that an interval of $k = 10$ provides optimal balance of resolution and speed for the systems we have tested.

To quantitatively characterize and reduce the dimensionality of the distributions, BARCODE computes the median skewness $S_{1,i}$, mode skewness $S_{2,i}$, and kurtosis $K_i$ for each evaluated frame $i$ (Fig. 2C), which are each metrics that describe the shape of the distributions relative to a Gaussian distribution. $S_{1,i}$ ($S_{2,i}$) is the difference between the median (mode) and mean (Fig. 2C), normalized by the standard deviation of the distribution, with positive values indicating a more pronounced tail of high intensity pixels and/or a peak at lower-than-expected pixel values. These features often correlate with bundling and aggregation which lead to denser (brighter) regions complemented with larger regions of minimal mass, which shift the distribution to having a more pronounced high-intensity tail and lower intensity mode, respectively. Because in skewed distributions, the median is typically closer to the mean than the mode is, the mode skewness $S_{1,i}$ is often a more sensitive skewness measure. However, the median skewness $S_{2,i}$ is less susceptible to artifacts of pixel saturation and photobleaching, which can cause modes to be at the highest and lowest pixel values, respectively. Kurtosis, computed as the fourth central moment of the distribution normalized by the squared variance, $K = \frac{\mu_4}{\sigma^4} - 3$, reports the extent to which pixel values are closer to ($K < 0$) or further from ($K > 0$) the mean than expected for a normal distribution (see SI, Section 1). Positive kurtosis values are indicative of de-mixing, coarsening and/or clustering while negative values indicate uniform and/or space-filling materials.

In general, if photobleaching were significant, one would expect the mean of the intensity distribution to decrease with time, shifting the overall distribution left towards zero. However, the shape of the distribution would likely not change substantially, since the underlying structures are not changing. By contrast, material restructuring due to aggregation, phase separation or bundling should change the distribution of mass, and should lead to changes in the distribution shape, which are detected via the kurtosis and skewness outputs of BARCODE. In the case of severe photobleaching, one might expect that eventually some fraction of low intensity pixels would become so dark that their intensity would not be detectable over the static pixel noise which arises from e.g., stray light, dark camera noise, etc. In this limit, we would expect the shape of the intensity distribution, and therefore the kurtosis or skewness, to likely change due to the inability to detect the full distribution (i.e., due to 'missing events'). To avoid this potential artifact, we flag videos with unusually low intensity and remove these from the analysis; saturated videos are also flagged (see SI Section 1).

Similar to island and void computations in the IB branch, BARCODE reports the maximum of each of these shape metrics across all evaluated frames, denoted as $S_1$, $S_2$ and $K$ (Fig. 2E), as well as the difference between the values in the initial and final $X\%$ of frames, $\Delta S_1$, $\Delta S_2$ and $\Delta K$, to determine the degree and type of restructuring. These 6 metrics accurately capture the differences between Video 1 and 2 and their time-dependent restructuring. In Video 1, we find that $S_{2,i}$ is positive and remains relatively constant, indicative of modest bundling seen in the example frames. $K_i$ is likewise positive for all frames but displays a much wider range of values over time and is substantially closer to zero at the end of the video ($\Delta K < 0$), indicating a more uniformly bundled material (i.e., a higher density of pixels near the mean).

The Optical Flow (OF) branch (Fig. 2D) outputs 4 BARCODE metrics by calculating velocity vectors for every pixel of every $k^{th}$ frame of a video, and averaging those vectors within each $p \times p$ window of the $m \times n$ image, resulting in a downsampled field of $m/p \times n/p$ velocity vectors for all $T/k$ frames of a video, where $k$ and $p$ are user-defined. From the vector fields, which we save as RDS, we compute, for each field $i$, the speed $v_i$, mean flow direction $\theta_i$, and directional spread $\sigma_{\theta,i}$, calculated as the circular standard deviation of flow directions across all vectors. Averaging each of these quantities across all fields from a given video provides the global averaged BARCODE metrics, $v$, $\theta$ and $\sigma_\theta$. We also compute the change in $v_i$ between the final and initial $X\%$ of vector fields $\Delta v$. To compute direction metrics we use bounds of $-\pi$ to $+\pi$ and take care in averaging the vector components to avoid issues with motion that is directed near the $\pm \pi$ boundary (see SI, Section 1). Together these BARCODE entries quantify the magnitude and direction of the material motion, and the extent to which it is directed or randomly oriented and speeding up or slowing down. For example, a material that exhibits fast directed flow in the vertical direction that slows down over time, will report a high $v$ value (fast), $\Delta v < 0$ (slowing down), $\theta \approx \pi/2$ (vertical direction) and $\sigma_\theta \approx 0$ (most vectors are oriented in the same direction). Steady but randomly oriented motion would result in $v > 0$ (the rate of random fluctuations), $\Delta v \approx 0$ (steady rate), $\theta \approx 0$ (all directions average to zero in the isotropic case) and $\sigma_\theta > 0$. Moreover, comparing the OF branch metrics between different channels of the same video reports the degree to which the different components of the material are moving together. OF analysis of the two example videos reveal starkly different dynamics (Fig. 2D). The Video 1 flow field has smaller vectors that have a wider range in orientations compared to Video 2, which can be seen quantitatively by $v_i$ being smaller and $\sigma_{\theta,i}$ being larger and more variable for Video 1 compared to Video 2. We also see that $v_i$ is relatively constant over time for Video 1 ($\Delta v \approx 0$), while it substantially decreases for Video 2 ($\Delta v < 0$). These distinct dynamics are captured in the different colors of the last 4 entries of their respective barcodes (Fig. 2E).

To benchmark BARCODE performance and demonstrate its ability to rapidly screen and discover properties of a broad range of non-equilibrium materials, we present the results of BARCODE analyses performed on microscopy data for four different materials that each

have different image sizes, resolution, durations, acquisition parameters, and dataset sizes.

## BARCODE accurately measures filament speeds and reveals emergent correlations in active cytoskeleton composites

We first analyze a published set of two-channel confocal videos of composites of entangled actin and microtubules, with and without crosslinkers, that undergo restructuring via kinesin and myosin motors acting on microtubules and actin, respectively (Fig. 3A)[20]. The dataset includes 48 videos of $T \approx 1000$ frames of size $m \times n = 512 \times 512$, each with two channels that separately visualize the actin and microtubules (Fig. 3B). Originally, the videos were analyzed using differential dynamic microscopy (DDM)[49,50], an established and powerful, yet labor-intensive, approach to show that all formulations exhibit ballistic motion with speeds spanning roughly three orders of magnitude. DDM analysis also revealed that the data could be categorized into three distinct dynamic classes: 'fast' directed flow, 'slow' isotropic restructuring or 'multi-mode' dynamics that manifested aspects of fast and slow behavior (Fig. 3B). Importantly, the dynamic class was not statistically correlated with the material formulation and the molecular underpinnings remained unclear.

BARCODE analysis of this dataset took ~9 min to complete, with no pre- or post-processing of the videos, compared to ~53 min required to complete the first step of DDM analysis, after which several rounds of subjective fitting and parameter choices requiring expertise are needed to determine dynamics. The resulting barcode matrices for the actin (Fig. 3C) and microtubule (Fig. 3D) channels also provide structural and orientational data for the complete dataset, which was not comprehensively analyzed in the original work[20]. The 4 OF entries are indistinguishable between the two channels, consistent with the nontrivial finding that actin and microtubules exhibited similar speeds across the entire formulation space[20]. The other branches show differences, indicating distinct and previously unreported structural features of actin and microtubule networks.

To first benchmark and validate BARCODE, we compare the speeds $v$ computed from BARCODE to the previously reported DDM-computed speeds for the entire dataset, color-coded by the nominal dynamical class (Fig. 3B, E). We find remarkably good agreement over two decades of speeds (Fig. 3E, SI Table S2). Multi-mode class data deviates from the equivalency line more than the other classes because the original DDM analysis resulted in two speeds, which were averaged here, while BARCODE computes a single speed for each condition. BARCODE also quantitatively captures other dynamical features of each class that were only qualitatively described in prior work. Specifically, prior particle image velocimetry (PIV) analysis demonstrated that a single exemplar video for each of the fast and slow classes exhibited unidirectional and randomly oriented motion, respectively. By evaluating correlations between speed $v$, directional spread $\sigma_\theta$, and flow direction $\theta$, BARCODE not only reproduces this result but demonstrates its applicability to all data in each class (Fig. 3F).

We next demonstrate the ability of BARCODE to rapidly discover structural properties and correlations. We observe substantially greater median and mode skewness changes, $\Delta S_1$ and $\Delta S_2$, for fast class data compared to slow and multi-mode (Fig. 3G), suggesting more pronounced restructuring. Moreover, the larger range in $\Delta S_2$ values compared to $\Delta S_1$ is a likely indicator of pixel saturation due to largescale aggregation, shifting the mode from a low intensity peak to the highest value (as shown in Fig. 2C). A powerful feature of BARCODE is its ability to directly correlate structural and dynamical features, such as the relationship between directional spread $\sigma_\theta$ and maximum median skewness $S_1$ (Fig. 3H). Here, we find clear partitioning between the different dynamic classes. Fast data exhibits minimal $\sigma_\theta$, indicative of directed motion; and a large spread in $S_1$, suggestive of wide-ranging structures. Conversely, the other classes generally exhibit higher $\sigma_\theta$

values, indicating more randomly oriented motion; and lower and less varied $S_1$ values, signifying minimal restructuring.

Using BARCODE's IB branch we compare the initial maximum and secondary maximum island areas, $I_{0,1}$ and $I_{0,2}$ (Fig. 3I), observing that slow and multi-mode composites exhibit predominantly a single large island, with $I_{0,2} < 5\%$ for all videos and $I_{0,1}$ values up to ~75%. Fast class composites have a comparatively broader spread in initial island sizes and generally smaller $I_{0,1}$ and larger $I_{0,2}$ values, suggesting that less connected and more heterogeneous networks may more readily facilitate fast flow. Additionally, microtubules generally have smaller island sizes compared to actin, which may reflect increased network heterogeneity and larger mesh sizes.

To corroborate this physical picture, we compare the connectivity and maximum void area for the different classes (Fig. 3J), finding that nearly all slow and multi-mode networks remain fully connected ($C \approx 1$) and most void areas are below 50%. Conversely, BARCODE reports a range of fractional ($C < 1$) connectivity values for the fast class networks as well as for microtubules, which also have larger void areas ($V > 50\%$) and smaller initial island areas compared to actin.

## BARCODE reveals universal features of active cytoskeleton materials

To further validate BARCODE and demonstrate its rapid and robust characterization capabilities, we analyze two additional datasets: a published study that examined the contractile behavior of myosin-driven crosslinked actin[10] (Fig. 4A), and previously unpublished data on kinesin-driven composites of actin and microtubules, with and without crosslinking (Fig. 4H). In the first, confocal microscopy videos with separate channels representing labelled myosin and actin were collected. Previously published image analysis performed on the myosin channels revealed three classes of contractile behavior: local, global, and critically connected (Fig. 4A). Local contraction was signified by actin and myosin forming small scale (~10 μm) uniformly distributed clusters, whereas global contraction led all components to condense into a single well-defined region. Critically connected networks exhibited distinct reconfiguration across lengthscales, with some regions condensing to large aggregates while others remained more homogeneously distributed (Fig. 4A).

To corroborate these general features of the different classes and discover their dynamical properties and structure-dynamics correlations, we use BARCODE to analyze both actin and myosin channels of the videos (Fig. 4B,C). Upon visual inspection of the barcodes, we find highly similar OF metrics between actin and myosin, suggesting that the two components are highly interacting and moving together. Both channels show larger maximum areas and larger changes for voids as compared to islands (i.e., $\Delta V > \Delta I$, $V > I$). Additionally, the actin channel shows a lower degree of connectivity and increased variability in skewness ($S_1, S_2$), suggesting more complex structure and reconfiguration, and consistent with the images in Fig. 4A. We next analyze correlations between the quantitative metrics output by BARCODE, color-coding the data by the previously identified contractile classes (Fig. 4D-G).

Comparing the speed $v$ to the speed change $\Delta v$ (Fig. 4D), we confirm that myosin and actin dynamics are highly correlated (i.e, similar $v$ and $\Delta v$ values). Locally contractile networks display relatively low speeds with minimal changes, consistent with small-scale local contractions. By contrast, both critically connected and globally contractile networks exhibit higher and more variable speeds, spanning over an order of magnitude in $v$, with faster speeds showing increased changes in dynamics (Fig. 4D inset).

To determine how the initial structure correlates with contractile class, we compare the initial maximum and secondary maximum island areas for actin and myosin (Fig. 4E). For actin, we find that locally contractile networks exhibit large $I_{0,1}$ values (>40%) and nearly zero $I_{0,2}$ values, similar to the slow class networks in Fig. 3I, and consistent

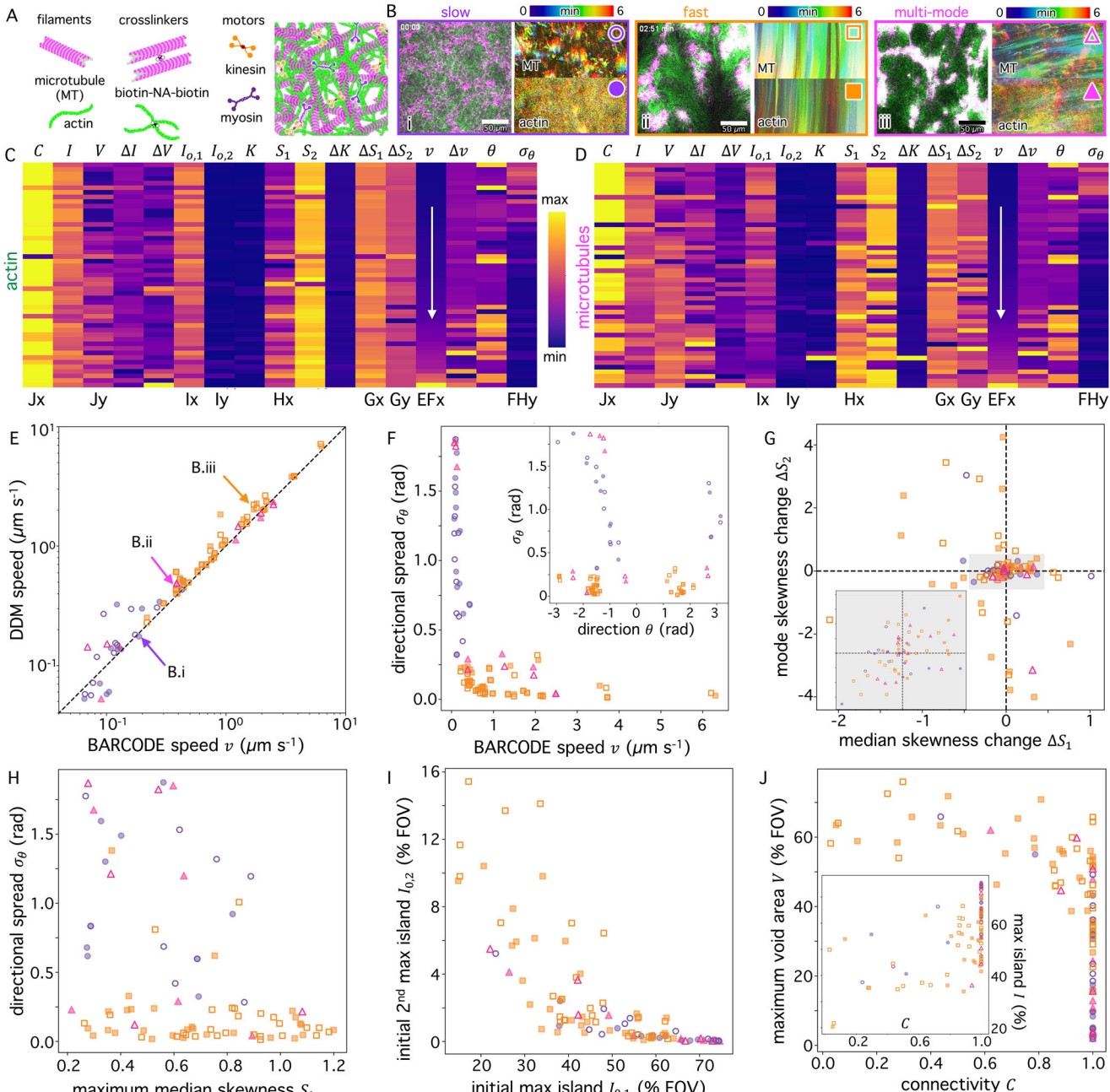

**Fig. 3 | BARCODE validation and discovery of emergent properties and relationships in motor-driven cytoskeletal composites. A** Cytoskeleton composite[20] components include actin (green), microtubules (magenta), crosslinkers (biotin-NeutrAvidin-biotin), and multimeric myosin (dark purple) and kinesin (orange) motors. **B** Multi-channel images of actin (green) and microtubules (magenta), and single-channel color-coded temporal projections, with features from each frame of a video colorized according to the time-color scale shown, for each of the previously identified dynamical classes [(*i*) slow: purple circles, (*ii*) fast: orange squares, (*iii*) multi-mode: magenta triangles] and filament types [actin: filled, microtubules:open], as used in plots E-J. Barcodes for actin (**C**) and microtubule (**D**) channels[20] ordered by increasing speed $v$. Letters below select columns denote the figure subpanel (E-J) and axis ($x$, $y$) where the metric is plotted. **E** Speed $v$ calculated with BARCODE plotted against speeds calculated using differential dynamic microscopy (DDM)[20], show excellent agreement over two orders of magnitude (see Table S2 for statistical analysis). The dashed equivalency line indicates where the two values are equal. Arrows indicate data points corresponding to the conditions shown in (**B**). **F** Correlation between speed $v$ and directional spread $\sigma_\theta$ shows faster speeds are correlated with directed motion (minimal $\sigma_\theta$). (Inset) Correlation between $\sigma_\theta$ and flow direction $\theta$ shows fast flows are directed along $\pm \frac{\pi}{2}$, while slow and multi-mode videos have no preferred direction. **G** Correlation between median skewness change $\Delta S_1$ and mode skewness change $\Delta S_2$, with dashed lines denoting no change. Grey inset is a zoom-in of the grey central region. **H** Correlation between flow directional spread $\sigma_\theta$ and maximum median skewness $S_1$ show that fast samples (orange) exhibit smaller directional spread than other classes, coupled with a wider range of larger skewness values. **I** Initial maximum island area $I_{0,1}$ versus initial secondary maximum island area $I_{0,2}$ shows slow and multi-mode composites are initially dominated by a single large island, while fast composites have broader distribution of initial sizes. **J** Correlation between connectivity $C$ and maximum void area $V$ show slow and multi-mode networks are largely connected while fast networks are often not connected. Inset: Connectivity $C$ versus maximum island area $I$.

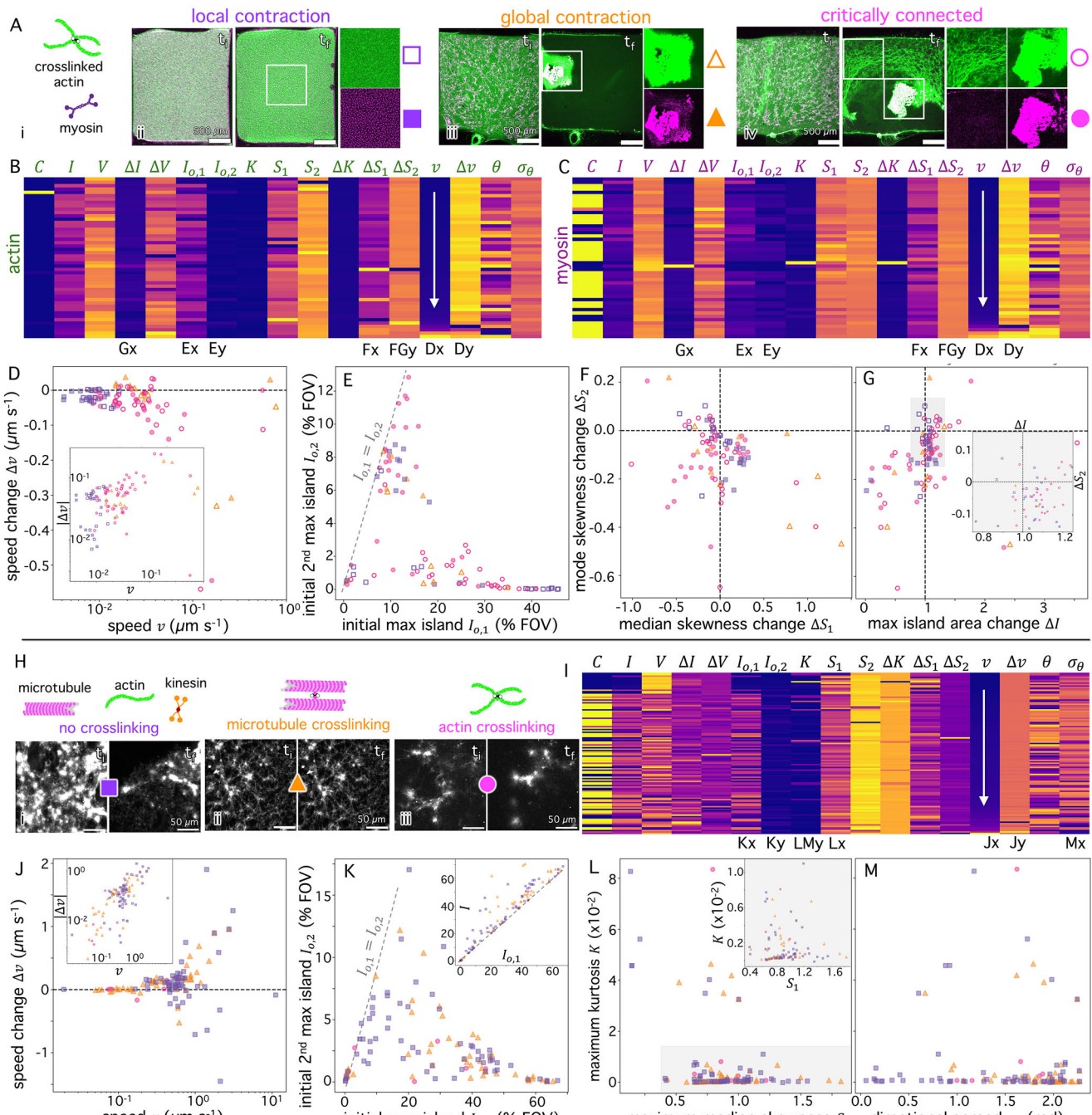

**Fig. 4 | BARCODE reveals universal dynamical and structural properties in diverse cytoskeletal networks. A** Networks of (*i*) crosslinked actin (green) and myosin (purple), previously examined using confocal fluorescence microscopy and classified as locally (*ii*, purple) or globally (*iii*, orange) contractile or critically connected (*iv*, magenta)[10]. Initial ($t_i$) and final ($t_f$) frames of representative videos of actin (green) and myosin (magenta) with single-channel zoom-ins of the boxed-in regions in $t_f$ frames. Colors and symbols for actin (open) and myosin (filled) in (*ii*) local (purple squares), (*iii*) global (orange triangles), and (*iv*) critically connected (magenta circles) networks are used in (**D–G**). Barcode matrix for actin (**B**) and myosin (**C**) channels[10], ordered by increasing speed $v$. Letters below metrics denote figure subpanel (**D–G**) and axis $(x, y)$ where they are plotted. **D** Speed $v$ versus speed change $\Delta v$. Inset: absolute value of $\Delta v$ vs $v$ on log scale. **E** Initial maximum island area $I_{0,1}$ versus initial secondary maximum island area $I_{0,2}$. Dashed line indicates $I_{0,1} = I_{0,2}$. Correlations between mode skewness change $\Delta S_2$ and (**F**) median skewness change $\Delta S_1$ and (**G**) maximum island area change $\Delta I$, with dashed lines denoting zero change. Grey inset is zoom-in of grey region. **H–M** BARCODE analysis of fluorescence confocal microscopy videos of networks comprising actin, microtubules and kinesin motors with and without crosslinking via biotin-NA-biotin. **H** Schematics of components (top) and initial and final frames (bottom) of example videos of fluorescent-labeled microtubules for composites with (*i*) no crosslinking: purple squares, (*ii*) microtubule crosslinking: orange triangles, (*iii*) actin crosslinking: magenta circles. **I** Barcode for all compositions ordered by increasing $v$. Letters below select metrics denote figure subpanels and axis $(x, y)$ where they are plotted. **J** Speed $v$ versus $\Delta v$. Inset: absolute value of $\Delta v$ vs $v$ on log scale. **K** Initial maximum island area $I_{0,1}$ versus initial secondary maximum island area $I_{0,2}$ with dashed line indicating $I_{0,1} = I_{0,2}$. Inset: $I_{0,1}$ versus maximum island area $I$ with the dashed line indicating their equality. **L, M** Correlations between maximum kurtosis $K$ and (**L**) maximum median skewness $\Delta S_2$ and (**M**) flow directional spread $\sigma_\theta$. Grey inset is zoom-in of grey region.

with prior reports[10,20]. Conversely, the globally contractile and critically connected networks have smaller $I_{0,1}$ and larger $I_{0,2}$ values, similar to the fast networks in Fig. 3I, and suggesting the presence of a larger number of structures of similar sizes. In contrast, the myosin data exhibit similar and generally correlated values of $I_{0,1}$ and $I_{0,2}$, with the critically connected networks having the highest $(I_{0,1}, I_{0,2})$ pairs, consistent with their expected clustering across different scales.

Finally, we relate restructuring to dynamics (Fig. 4F,G), comparing changes in mode skewness $\Delta S_2$ to changes in median skewness $\Delta S_1$ (Fig. 4F) and island area $\Delta I$ (Fig. 4G). We observe clear distinctions between the different contractile classes and between actin and myosin. We find that actin exhibits larger skewness changes compared to myosin, as visually indicated by the barcode (Fig. 4B,C), and global and critical contractile networks exhibit larger skewness changes compared to local contraction data, as expected. Interestingly, critical and global data have mostly negative and positive median skewness changes $\Delta S_1$, respectively, indicating large-scale aggregation (brighter pixels) coupled with more network-poor (darker) regions for global data as compared to varied or multiple aggregates (brighter pixels) coupled with sustained network-rich regions (fewer dark pixels) for critically connected networks. These results highlight the power of BARCODE to reveal complex behaviors and correlations in active systems from a remarkably thrifty set of metrics produced in min.

Thus far, we have presented analysis of previously published and well-vetted data to benchmark BARCODE, revealing insights and emergent similarities between two different cytoskeletal materials. To explore potential universality in response, and further demonstrate BARCODE capabilities, we next analyze previously unpublished data on kinesin-driven composites of actin and microtubules, with and without crosslinking of actin or microtubules (Fig. 4H). The system components are similar to those shown in Fig. 3A, but with a lower ratio of microtubules to actin, higher kinesin concentration, and no myosin (see SI Section 2). The actin is also not labeled in this set of 109 videos, so our analysis focuses solely on the structure and dynamics of the microtubule network. Comparing the barcodes for this system (Fig. 4I) to those of the actomyosin system (Fig. 4B), we find similar OF parameters, except $\Delta v$ appears higher for the kinesin-driven composites, while we observe much more variability in IB metrics. To further investigate similarities and differences between the different cytoskeleton systems, we evaluate similar correlations as in Figs. 3 and 4D-G, color-coding the data by the type of crosslinking: none, microtubule-microtubule, or actin-actin.

We again find that slower speeds correlate with minimal speed changes ($\Delta v \approx 0$) (Fig. 4J), and increasing speed generally correlates with increasing speed change $|\Delta v|$ (Fig. 4J inset). However, in comparison to actomyosin network dynamics, the speeds are generally an order of magnitude higher and dynamics appear to speed up ($\Delta v > 0$) rather slow down. Moreover, uncrosslinked networks generally exhibit faster speeds and greater speed changes compared to crosslinked networks, and crosslinked microtubule networks exhibit the slowest speeds likely because crosslinking provides more rigidity and connectivity thereby slowing contraction. Increased connectivity should lead to larger differences between the initial maximum and secondary maximum island areas, since connected networks should have FOV-spanning islands, which we observe (Fig. 4K). Most of the crosslinked data lies below the equivalency line, indicating $I_{0,1} > I_{0,2}$, with a roughly inverse relationship between $I_{0,1}$ and $I_{0,2}$, similar to trends in Fig. 3I and the actin data in Fig. 4E. By contrast, the uncrosslinked network data falls near the equivalency line, indicating an ensemble of similarly sized clusters. Contraction is further confirmed by comparing $I_{0,1}$ with the maximum island area $I$ (Fig. 4K inset), which shows that most paired $I_{0,1}$ and $I$ values are similar, suggesting that islands are generally shrinking (i.e., contracting) over time.

To provide further insight into the network restructuring, we evaluate correlations between the maximum kurtosis $K$ and the

maximum median skewness $S_1$ (Fig. 4L) and flow directional spread $\sigma_\theta$ (Fig. 4M). We find that $K$ and $S_1$ are generally inversely related and that most crosslinked networks exhibit lower kurtosis and higher skewness than uncrosslinked networks, which we attribute to increased connectivity and bundling, respectively. Specifically, lower $K$ values and higher $S_1$ values indicate increased global structural homogeneity and increased local mass density, respectively. Coupling these structural distinctions with dynamical features, we observe that uncrosslinked networks exhibit a wider spread in $\sigma_\theta$ values that extend to much smaller values than for crosslinked networks (Fig. 4M), indicating more variable types of motion and a higher propensity for directed motion in the uncrosslinked case. This behavior is likely enabled by the reduced connectivity, increased structural heterogeneity, and faster speeds that uncrosslinked networks exhibit compared to crosslinked networks. Together, these results demonstrate BARCODE's ability to quantify and correlate a host of structural and dynamic parameters for different active cytoskeleton network formulations, conditions, acquisition parameters, and spatiotemporal scales to reveal universal behaviors and emergent properties in this foundational class of active matter.

## BARCODE characterizes dynamics and structure of active cell monolayers

To further demonstrate the broad applicability of BARCODE, we next analyze two datasets of dynamic cell monolayers. In each, large FOV (>1 mm) time-lapse videos are captured over the course of days (69-89 hrs) and the nuclei and cytoplasm are imaged separately, providing two channels for analysis. We first evaluate a previously published dataset[34] of videos of weakly interacting spindle-shaped human dermal fibroblasts (hdFs) (Fig. 5A) at two cell densities. The barcodes for each channel of the 72 videos (Fig. 5B) show both expected and nontrivial properties. For example, the entire set of nucleus data, which primarily contains isolated punctate objects that are not expected to significantly change shape or size, has essentially no connectivity, very small maximum and initial island sizes, and large voids. Conversely, the cytoplasm channel has more instances of positive connectivity along with larger maximum island areas and smaller voids. Despite clear structural differences, the dynamic OF metrics ($v, \Delta v, \theta, \sigma_\theta$) display similar trends between the channels, as expected since the two components comprise the same ensemble of cells. To more closely examine the effects of cell density, we evaluate similar correlations as in Figs. 3 and 4.

Evaluating the speed and its change for each channel and cell density (Fig. 5C), we observe modest slowing for nearly all data, perhaps indicative of jamming as the cell number increases over time due to cell division, in line with the observation that the higher cell density data exhibits generally slower speeds. The cytoplasmic signal is generally slower than the nuclear signal, which we attribute to the more complex shape and intensity changes of the cytoplasm. Comparing the initial maximum and secondary maximum island areas, we find most of the data falling near the $I_{0,1} \approx I_{0,2}$ equivalency line. Moreover, the nucleus data is all tightly clustered around $I_{0,1} \approx I_{0,2} \approx 1\%$ which is equivalent to ~2000 $\mu m^2$, roughly equivalent to the 2D projected area of the nucleus (Fig. 5A). The cytoplasm data displays more variation in island area and greater deviation from the equivalency line (i.e., $I_{0,1} > I_{0,2}$), indicating closely packed cells that register as larger multi-cell islands.

Turning to structural changes, quantified by $\Delta S_2$ and $\Delta V$ (Fig. 5E), we observe minimal changes in the nucleus channel, as expected, since they are not changing in size, shape or concentration. The cytoplasm data shows more variations in both metrics, reflecting changing cell shapes as well as clustering over time. Moreover, $\Delta S_2$ is universally negative, indicative of increased uniformity and reduced spread in pixel intensities, which can be observed in Fig. 5A. Interestingly, we find that the maximum void areas decrease for low density cases and

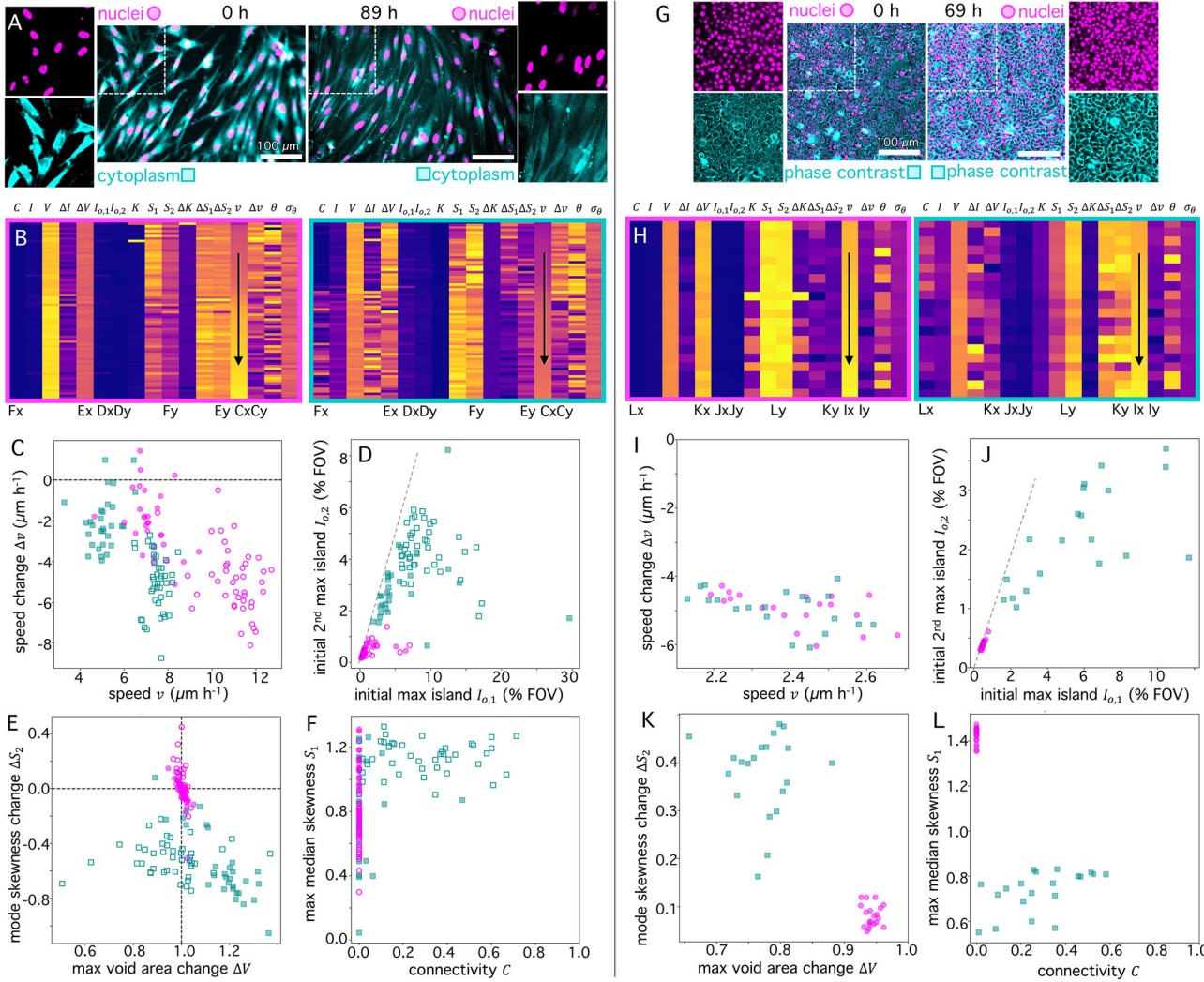

**Fig. 5 | BARCODE provides high-throughput characterization of dynamic cell monolayers. A–F** BARCODE analysis of epifluorescence microscopy videos of monolayers of human dermal fibroblast (hdF) cells with spectrally-distinct labeled nuclei (magenta, circles) and cytoplasm (cyan, squares) and two different cell concentrations [300 mm⁻² (open), 450 mm⁻² (filled)] as previously reported[34]. Scale bar is 100 μm for all images. **A** Initial (0 h) and final (89 h) frames of representative multi-channel video of low-density monolayers with single-channel zoom-ins of the boxed-in regions in each frame. **B** Barcode matrix for the nuclei (left, magenta border) and cytoplasm (right, cyan border) channels of the source videos[34] ordered by increasing $v$. Letters below select columns denote the figure subpanel (**C–F**) and axis $(x, y)$ where the metric is plotted. **C** Correlation between speed $v$ and $\Delta v$ shows general slowing of dynamics and faster motion for lower concentration monolayers. **D** Correlation between initial maximum $I_{0,1}$ and secondary maximum $I_{0,2}$ island areas with dashed line indicating $I_{0,1} = I_{0,2}$. **E** Mode skewness change $\Delta S_2$ versus void area change $\Delta V$ with dashed lines denoting no change. **F** Maximum

median skewness $S_1$ versus connectivity $C$. **G–L** BARCODE analysis of epifluorescence and phase contrast microscopy videos of monolayers of jamming MCF10A breast epithelial cells prepared as described[37,38]. Color and symbol keys, correlation plots and RDS mirror those in (**A–F**). **G** Initial (0 min) and final (69 h) frames of representative multi-channel video of monolayers with single-channel zoom-ins of the boxed-in regions in each frame. **H** Barcode matrix for the nuclei and phase contrast channels, ordered by increasing $v$. **I** Correlation between $v$ and $\Delta v$ shows general slowing of dynamics and similar speeds for nuclei and phase contrast channels, the latter of which represents the cell boundary and cytoplasm. **J** Correlation between $I_{0,1}$ and $I_{0,2}$ with dashed line indicating $I_{0,1} = I_{0,2}$. **K** Mode skewness change $\Delta S_2$ versus maximum void area change $\Delta V$ shows voids universally shrink and skewness increases, an effect that is amplified for data obtained from phase contrast imaging. **L** Maximum median skewness $S_1$ versus connectivity $C$.

increase for high densities. This suggests that at higher densities cell cluster into large connected multi-cell regions producing larger void areas, while at lower density, cells remain more dispersed, thus producing smaller voids. This physical picture is corroborated by Fig. 5F which shows that lower cell densities result in higher connectivity $C$ and generally higher median skewness $S_1$, indicating a more connected meshwork of cells at low densities compared to large multi-cell aggregates that come with larger voids and more dark pixels.

To assess the relevance of these findings to other cell systems, we also analyze videos of jamming MCF10A breast epithelial cells (Fig. 5G-L). Here, whole cells and their GFP-labeled nuclei are imaged using phase contrast and epifluorescence, respectively (Fig. 5G). Examining

the barcode for each channel (Fig. 5H), we find similarities with the hdFs (Fig. 5B), including minimal connectivity and island areas for the nucleus data and similar OF metrics between the two channels. Evaluating the same correlations as in Fig. 5C-F, we find the both cell types exhibit generally decreasing speeds (i.e., $\Delta v < 0$) (Fig. 5C,I) and similar $I_{0,1}$ vs $I_{0,2}$ correlations (Fig. 5D,J). However, the spread in $\Delta v$ and $v$ values is less for the MCF10As (Fig. 5I) compared to hdFs (Fig. 5C), suggesting more ordered and correlated motion within epithelia as compared to more motile fibroblasts. Moreover, the mode skewness change is positive ($\Delta S_2 > 0$) for epithelial cells (Fig. 5K) in contrast to hdFs and consistent with increased contrast and sharpness of final versus initial images (Fig. 5G,K), opposite to hdFs. Finally, we observe

zero connectivity ($C = 0$) (Fig. 5F,L) and minimal void area change ($\Delta V \approx 1$) (Fig. 5E,K) for the nucleus channel and a wide range of connectivity values among the cell boundaries (Fig. 5F,L). Together, our results demonstrate the power of BARCODE to reveal convergent properties, dissect differences between disparate materials systems and accurately characterize material properties using distinct imaging modalities.

## Discussion

We present BARCODE, an HTP analysis platform that rapidly extracts a unique low-dimensional fingerprint from large and complex microscopy datasets of active materials. Each barcode output represents a readily accessible visual readout of 17 quantitative features, providing a concise and standardized summary of the system's structural and dynamic behavior and enabling rapid identification and comparison of properties among different experiments, formulations and conditions. In generating each barcode, the platform also produces and archives standardized reduced data structures (RDS), including time series of binarized frames, intensity distributions, and optical flow fields which are calculated for a down-sampled selection of frames throughout the video. The RDS facilitate down-stream analysis and enable users to perform deep-dives into datasets that exhibit performance targets, irregularities or emergent outputs.

BARCODE offers several advantages for analysis of large microscopy-based datasets, as demonstrated by our analysis of multiple distinct active matter systems here. BARCODE is effective and efficient in reproducing known material properties and discovering new features and formulation-structure-dynamics relationships in a material-agnostic manner. We have established BARCODE's abilities to analyze confocal fluorescence, epifluorescence and phase contrast videos ranging from 6 min to 89 h, 5 to 1178 frames, and with dimensions of 200 μm to 2 mm and 256 to 6318 pixels. The barcodes produced for each data set included multiple video channels, with individual file sizes as large as 1.95 GB and folder sizes as large as 40 GB. For all datasets, BARCODE analysis was completed in a range between 10 min for a 12 GB dataset to ~75 min for a 40 GB dataset.

BARCODE is expressly designed to be a high throughput screening tool, rather than an intensive data analysis platform, to rapidly extract average and reduced-dimensionality metrics for coarse characterization to compare datasets, guide optimization and discover emergence. This data reduction necessarily leads to some information loss. To increase processing speed and ensure the metrics are applicable to a broad range of materials, we chose to reduce spatial and temporal resolution (downsampling frames and pixels by $k$ and $p$) in calculating the RDS, and to focus on extremal quantities and changes thereof. In general, the metrics that report changes in quantities assess property differences between the beginning and end of each video, so transient changes may not be captured in the lowest dimension barcode matrix. However, the full time-dependent information is saved in the RDS and users can choose to perform advanced analysis on RDS to assess time-dependence and discern more subtle changes in material structure and dynamics.

To ensure broad accessibility, we designed BARCODE to require minimal subjective inputs or training, and to be executed with a user-friendly GUI, in contrast to algorithmically-intensive methods such as Fourier image analysis and particle-tracking methods, allowing for rapid analysis and broad use of the platform by beginners and experts alike[49,51–54]. Our improved ability to analyze and share data is an important step towards democratizing materials discovery across disparate working groups with varied expertise.

BARCODE is also highly adaptable: branches and metrics can be easily added or removed, providing flexibility to users and enabling the software to meet the future needs of the community. Planned expansions of BARCODE will include additional branches and metrics that quantify diffusive behavior and mechanical properties, including two-point correlation functions, as well as adding user-identified improvements and user-generated extensions in the future. We are also currently investigating the use of machine learning for data classification and to enable data-driven predictions of material behavior. Ultimately, the rapid read-outs BARCODE provides may also enable 'on-the-fly' training of machine learning tools for active material design and optimization, thereby unlocking the capabilities of complex multiphase multiscale soft materials in fit-for-purpose applications.

## Methods

### Preparation and imaging of active cytoskeleton networks

Data presented in Figs. 2 and 3 were generated from previously published source videos[20]. Actomyosin data presented in Fig. 4A-G were generated from previously published source videos[10]. The cytoskeletal networks and video data presented in Fig. 4H-M were generated as follows.

**Proteins.** All proteins were purchased as lyophilized powder from Cytoskeleton, Inc, reconstituted, flash-frozen into single-use aliquots, and stored at −80 °C until use. Rabbit skeletal actin monomers (AKL99) and biotin-actin monomers (AB07) were reconstituted in G-buffer (2.0 mM Tris (pH 8), 0.2 mM ATP, 0.5 mM DTT, 0.1 mM CaCl₂). Porcine brain tubulin dimers (T240), HiLyte 647-labeled tubulin dimers (TL670M) and biotin-tubulin dimers (T333P) were reconstituted in PEM-100 buffer (100 mM PIPES (pH 6.8), 2 mM MgCl₂, and 2 mM EGTA).

Biotinylated kinesin-401[55,56] was expressed, as described previously[20], in Rosetta (DE3)pLysS competent *E. coli* (ThermoFisher) grown on selective media plates for 16-18 h at 37 °C. Fifteen colonies were added to a 5 mL starter culture of selective LB media and grown for 2 h at 37 °C/250 rpm before adding to 400 mL of selective LB media. Cells are grown at 37 °C/250 rpm to OD 0.6-0.9 at 600 nm, then induced at 20 °C/250 rpm for 18 h with 1 mM Isopropyl β-D-1-thiogalactopyranoside (IPTG), and pelleted at 5000 rpm for 10 min at 4 °C before being frozen at −80 °C for 1 h. Cells were lysed in lysis binding buffer (50 mM PIPES, 4 mM MgCl₂, 20 mM imidazole, 10 mM β-mercaptoethanol, 50 μM ATP, one protease inhibitor tablet per 10 mL, 1.1 mg/mL PMSF, 1.1 mg/mL lysozyme) via sonication for 3 mins, pulsing every 20 s, then pelleted for 30 mins at 40,000 x g at 4 °C, filtered through a 0.22 μM filter, and incubated with 1 mL nickel (Ni-NTA) agarose beads (Qiagen) for 2 h on a rocker at 4 °C. The lysate/bead mixture was passed through a chromatography column then washed with 15 mL buffer (50 mM PIPES, 4 mM MgCl₂, 20 mM imidazole, 10 mM β-mercaptoethanol, 50 μM ATP, one protease inhibitor tablet per 10 mL) before 1 mL fractions were eluted in (50 mM PIPES, 4 mM MgCl₂, 20 mM imidazole, 10 mM β-mercaptoethanol, 50 μM ATP, one protease inhibitor tablet per 10 mL, 2 mM DTT, 0.05 mM ATP). An elution dot blot was performed to assess the most concentrated fraction which was run through a 40kD MWCO desalting column for buffer exchange with PEM-100 with 0.1 mM ATP, then mixed with 60% sucrose for a final concentration of 10% sucrose before being aliquoted and flash-frozen into single-use aliquots.

To prepare force-generating kinesin clusters, kinesin-401 dimers were incubated with NeutrAvidin (ThermoFisher) at a 1.7:1 ratio in PEM-100 to a final concentration of 10 μM supplemented with 4 mM DTT for 30 min at 4 °C. Clusters were prepared fresh and used within 24 h.

For composites that incorporate actin or microtubule cross-linking, actin:actin or microtubule:microtubule crosslinked complexes were prepared by combining biotinylated actin monomers or tubulin dimers with NeutrAvidin and free biotin at a ratio of 2:2:1 protein:free biotin:NeutrAvidin and incubating for 90 min[57].

**Cytoskeletal network preparation.** Actin-microtubule composites were prepared by polymerizing a mixture of 5.22 μM actin monomers, 6.06 μM tubulin dimers, 0.32 μM HiLyte 647-labeled tubulin dimers,

5.22 μM phalloidin (Invitrogen) and 5 μM Taxol (Sigma) in PEM-100, supplemented with 0.1% Tween, 4 mM ATP, and 4 mM GTP. For crosslinked composites, a portion of either actin monomers or tubulin dimers was replaced with equivalent crosslinker complexes to achieve the same overall actin and tubulin concentrations and cross-linker:protein ratios of $R_A = 0.02$ for actin or $R_T = 0.005$ for tubulin. Composites were polymerized in the dark for 1 h at 37 °C.

To prepare active cytoskeletal composites, 5 μL of the polymerized actin-microtubule composite was combined with the following to a 9 μL final volume: oxygen scavenging system [45 μg/mL glucose, 0.005% β-mercaptoethanol, 43 μg/mL glucose oxidase, 7 μg/mL catalase, 2 mM Trolox (Sigma)] and ATP-regeneration system [26.7 mM phosphoenol pyruvate (Beantown Chemical, 129745) and pyruvate kinase/lactate dehydrogenase (Sigma, P-029)]. Finally, 1 μL of kinesin clusters was added to reach a final concentration of 1 μM, followed by gentle mixing of the sample by pipetting up and down.

**Sample chambers**. Sample chambers with a volume of ~10 μL were made by placing two strips of parafilm between a No. 1 glass coverslip and microscope slide, followed by heating to fuse them together. To prevent surface adsorption of proteins, the chambers were filled with a 1 mM solution of BSA (bovine serum albumin, Sigma) and incubated for 10 min, after which the solution was flushed out with compressed air. The prepared cytoskeletal network solution was then loaded into the chamber and the open ends were sealed with UV-curable glue.

**Imaging**. Imaging of HiLyte647-labeled microtubules was performed using a Nikon A1R laser scanning confocal microscope with a 60× oil-immersion objective (Nikon) and a 640 nm laser with $624 \pm 20$ nm / $692 \pm 20$ nm excitation/emission filters. Time-series (videos) of 256 × 256 square-pixel (213 μm × 213 μm) images were collected at 1.33 fps for a minimum of 400 frames (300 s). Imaging began 5 min after the addition of kinesin motors to the sample in the middle plane of the ~100 μm thick sample chamber. Each subsequent video was recorded in a different field of view laterally translated by at least 500 μm. Imaging continued until restructuring or motion was no longer visible (~60-120 min). 5-10 videos were collected for each sample. Each data point shown in Fig. 4 corresponds to a single video. The 55, 7 and 47 data points for the uncrosslinked, actin crosslinked and microtubule crosslinked data, respectively are from 8, 2 and 6 independent replicates, which sufficiently capture the range of material dynamics.

**Preparation and imaging of cell monolayers**
Data presented in Fig. 5A-F were generated from source videos[34] of human dermal fibroblasts (ATCC PCS-201-010), which we further processed as described below. Video data presented in Fig. 5G-J, from experiments similar to previous reports[37,38], were graciously shared by Jasmin Di Franco and Roberto Cerbino (University of Vienna, Austria) and prepared as described below.

**Cell culture**. MCF10A cells (kind gift of J. S. Brugge, Department of Cell Biology, Harvard Medical School, Boston, USA) were cultured in Dulbecco's Modified Eagle Medium: Nutrient Mixture F-12 supplement with Glutamine (DMEM/F12 GlutaMax) medium (Gibco), supplemented with 5% horse serum (Biowest), 0.5 mg/mL hydrocortisone (Sigma-Aldrich), 100 ng/mL cholera toxin (Sigma-Aldrich), 10 μg/mL insulin (Roche), 1% Penicillin-Streptomycin (HyClone), and 20 ng/mL EGF (Peprotech), the latter being added directly to the culture plates. Cells were maintained at 37 °C in a humidified atmosphere with 5% $CO_2$. Cell identity was verified by fingerprinting by the IFOM (Milan) cell culture facility, and cells were routinely tested for Mycoplasma contamination. Stable expression of GFP-H2B was achieved by lentiviral infection of MCF10A cells with pBABE-puro-GFP-H2B vectors to enable nuclear labeling.

**Cell jamming assay**. Cells were seeded into six-well plates at a density of $1.5 \times 10^6$ cells per well in complete medium and cultured to form a uniform monolayer (~24 h). Prior to imaging, the cell monolayer was carefully washed with 1X Dulbecco's Phosphate Buffered Saline (DPBS) to remove floating cells, and the medium was refreshed. Time-lapse images of size 1024 × 1024 square-pixels (1331 μm × 1331 μm) were captured every 5 min over a 72 h period using a Leica Thunder inverted microscope equipped with a 10× objective, both in phase contrast, to image the cell walls and cytoplasm, as well as fluorescence, to image the GFP-labeled nuclei. The assay was conducted in an environmental microscope incubator set to 37 °C with 5% $CO_2$ perfusion. Data presented in Fig. 5G-J are from 20 videos from 5 independent samples, each containing a fluorescence and phase contrast channel, which sufficiently capture the range of material dynamics. For BARCODE analysis, we divided each video channel into four 512 × 512 square-pixel images to increase statistics.

**Figure 5A-F video post-processing**. Videos[34] of size 6318 × 3546 square-pixels (3702 μm × 2078 μm) and 5088 × 2332 square-pixels (2982 μm × 1367 μm) were divided into smaller FOVs to increase statistics and remove spurious dark spots and sample borders that impact image analysis. Specifically, the data is from 72 videos from 2 independent samples, each containing two channels, which sufficiently capture the range of material dynamics and represent the full published dataset for these cell densities on isotropic subtrates. For BARCODE analysis, we divided each video channel into 30-42 video tiles with tile size of 960 × 608 square-pixels (562 μm × 35 μm).

**Reporting summary**
Further information on research design is available in the Nature Portfolio Reporting Summary linked to this article.

## Data availability
The source data for all plots shown in Figs. 2–5 have been deposited in the Dryad repository and can be accessed at: https://doi.org/10.5061/dryad.pc866t235. Source data are provided with this paper.

## Code availability
BARCODE and supporting documentation are available on the GitHub repository[58] at https://github.com/softmatterdb/barcode and can be accessed at: https://doi.org/10.5281/zenodo.17585069.

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

## Acknowledgements

BARCODE was developed through Hackathon events held at University of California, Santa Barbara and supported by the US National Science Foundation (NSF) Designing Materials to Revolutionize and Engineer our Future (DMREF) program with contributions from the following participants: Jonathan Michel, Mengyang Gu, Prashali Chauhan, Laura Morocho, Nimisha Krishnan, Anindya Chowdhury, Lauren Melcher, JJ Siu, Gregor Leech, Mehrzad Sasanpour, and Karthik Peddireddy. We acknowledge funding from the US National Science Foundation DMREF program through following grants: NSF DMR-2119663 (to RMRA), NSF DMR-2118403 (to JLR), NSF DMR-2118449 (to MD), NSF DMR-2118497 (to MTV), Research Corporation for Science Advancement award no. CS-PBP-2023-019 (to RMRA, MH), Arnold and Mabel Beckman Foundation Beckman Scholars Program (to RMRA, KM) and the NSF BioPACIFIC Materials Innovation Platform NSF DMR-1933487 for personnel support (QC) and access to research infrastructure. We thank Christopher Dunham, BioPACIFIC MIP, Emmie Kao and Christopher Tao for helpful discussions and preliminary research development, and Eric Feng for assistance in updating the graphical user interface. We thank Jose Alvarado, Gijsje Koenderink, and Yimin Luo for providing data[10,34] and for helpful discussions. We thank Roberto Cerbino and Jasmin Di Franco for providing unpublished data and for helpful discussions.

## Author contributions

M.T.V. and R.M.R.A. conceived and designed the research. Q.C., A.S., A.D., K.M., M.H., K.T., R.J.M., R.M.R.A. and M.T.V. designed BARCODE algorithms, pipeline, and documentation. K.M. and M.H. performed experimental work and curated data. Q.C., A.S., A.D., R.M.R.A. and M.T.V. analyzed data. Q.C., A.S., R.M.R.A., and M.T.V. prepared figures and wrote the manuscript. M.D., R.J.M., R.M.R.A., and M.T.V. supervised the research. All authors interpreted data and edited the manuscript.

## Competing interests

The authors declare no competing interests.
