## [Transparent Peer Review file · Nature Communications]

BARCODE: high throughput screening and analysis of soft active materials

Corresponding Author: Professor Megan Valentine

Version 0:

Reviewer comments:

Reviewer #1

(Remarks to the Author)

The main challenge addressed in the article, as I see it, is managing large-scale data—both in terms of enabling effective collaboration (a term I personally prefer over "democratization") and in extracting meaningful insights despite limited computational capacity due to data volume. This is indeed a critical problem, and addressing it can significantly advance scientific research.

However, my primary concern lies in the selection of parameters. Specifically, I question whether using only 17 parameters is sufficient to capture the complexity of such a large dataset. One potential solution would be to start with a broader set of features and then apply dimensionality reduction techniques. If a larger set of features were used, alternative methods of visualization might also be necessary.

Additionally, I would suggest outlining potential next steps in leveraging big data and machine learning techniques. One such step could be the automatic extraction of relevant features or parameters.

In summary, I believe the article could benefit from better contextualization for readers who are less familiar with data science. It should also give more critical attention to the implications of representing a massive dataset with a relatively small number of parameters.

(Remarks on code availability)

Not provided

Reviewer #2

(Remarks to the Author)

This manuscript presents a reasonable straightforward method called BARCODE to characterize several interesting properties of microscopy videos, which can then be used to compare videos from different experiments. The method is flexible as regards to the number of images and their exact dimensions, which is great. Several example data sets are used as demonstrations that the method is powerful. I think this is publishable after the authors clarify some points.

1. I'm most concerned about the Image Binarization branch. There is one user-defined threshold ('offset' defined in supplemental section 1). It seems that the results will be quite sensitive to this choice; that for example, the connectivity depends on this choice. Of course, BARCODE has far fewer parameters than many competing methods, so this one user-defined parameter is not a fatal flaw. Nonetheless, the authors should comment on how users might pick this threshold, and how sensitive the results can be on this choice. For example, which of the results presented in Figs. 2-5 would change significantly based on the choice of the threshold? The author list is long, so if different authors independently looked at the images and chose thresholds, how much variability occurs? Again: this is not a huge problem, but I suspect that different data will have different sensitivity to these choices, and different people might make different choices, so the ability to compare across data sets and between different research groups would potentially be diminished.

2. For the Intensity Distribution branch, how does photobleaching affect the results? In the paragraph on page 4 introducing the ID branch, it interprets the decrease of the mean intensity in Fig. 2C as due to the formation of aggregates. In some of my data sets, it would be due to photobleaching, which is not the same science as aggregation. How much is this a problem?

3. I'm also wondering how much uneven illumination would affect both IB and ID questions. Some days my fluorescent arc lamp is better aligned than others.

4. For the flow direction theta, it took me a while to understand why this is useful. The point is that if one has a two channel image, if both channels indicate the same flow direction, then the two components being visualized are co-moving. That's great. This should be explained, because if I'm just comparing across multiple experiments, the particular angle theta seems irrelevant. One could change it just by rotating the sample or camera, presumably.

5. But more importantly for theta, how does the software handle the definition of theta as between 0 and 2 pi? That is, suppose the motion is close to theta = 0. Fluctuations of the direction might mean that sometimes the direction is plus or minus a few degrees; but negative angles will be close to 2pi. The average of 0 and 2pi is pi, the opposite direction of the actual motion. Moreover, the standard deviation (Eqn. 27) will be much larger than it should be. How is this handled for both computing mean theta and computing the standard deviation?

6. Related to this, Fig. 3F(inset) caption says "fast flows are directed along plus/minus pi." This seems like it must be an artifact -- it depends on the camera orientation -- why would plus or minus pi be special? Although looking closely at the inset, I'm not seeing how "plus/minus pi" is deduced. pi is 3.14 and the horizontal axis of the inset ranges from -2 to +2, so does not include plus or minus pi at all. I am really confused.

7. In Fig. 3B caption it talks about "temporal colormaps." I do not understand what these are, please define.

8. The supplemental materials are nice, and generally tell exactly how to compute the various quantities. The exception is the kurtosis: please give a formula for how the kurtosis is calculated. The reason I'm asking is that I normally think of skewness as calculated based on moments of the distribution, but BARCODE is instead using "median skewness" and "mode skewness" which are not based on moments. That's fine, but it makes me wonder if the kurtosis is calculated based on moments. That is, is the kurtosis the fourth standardized moment defined on Wikipedia (<https://en.wikipedia.org/wiki/Kurtosis>)? And why not use the third standardized moment for defining skewness (<https://en.wikipedia.org/wiki/Skewness>)? I don't feel strongly about any of this, other than to just simply include an explicit formula for kurtosis in the supplemental materials.

(Remarks on code availability)

Reviewer #3

(Remarks to the Author)

This article describes a new analysis technique called BARCODE. This technique takes large data sets of videos and is able to process it quickly into many different variables that are then tested to see whether they are correlated. The text describes several examples from already analyzed data sets to show the utility of the technique and also shows how BARCODE can be used on new data to find new correlations between variables. This is a really interesting technique that could be impactful especially with the amount of data that can be collected steadily increasing. The manuscript is written in a way that makes it somewhat hard to understand the technique. As mentioned below, I think using less examples and treating the development of the technique as part of the results and providing a more thorough description in the main text would greatly improve the manuscript. I read the manuscript first and then the SI and it was not until I read the SI that I really started to understand how BARCODE worked. Not all details in the SI need to be added to the main text, but some descriptions of the technique do need to be moved to the main text. This can be done by moving an example or two to the SI. Due to this and other comments provided below, major revisions are required prior to publication in Nature Communications.

Comments, questions and concerns:

1. The three channels chosen for independent analysis seem to be limiting to the technique. What is the rationale for choosing these three channels and what kind of information is excluded by choosing these?
2. A colorbar should be provided for Fig. 1E, even if it is just qualitative.
3. There seems to be some redundancy in the first paragraph of the Results, the 3 image processing tools and the reduced data structure, I think are the same but are both listed a sentence apart, are they the same and is this redundant?
4. How does reducing the data change the accuracy of the results?
5. It seems that time-dependent data is lost in the data reduction, is that correct? If this is correct, how does this limit the method?
6. What are the arrows in Figure 3E?
7. If the changes in the images are not in a single direction, would this method work?
8. It is hard to follow the examples provided since they are described in terms of the variables and not well connected to the physical results in the movies. The actin-myosin example did a better job of connecting the variables to the physical changes in the videos but some discussion was still very hard to follow. It would help if the variables being described are related to what they physically represent in the videos, that would aid in the readability of the manuscript and better illustrate

the power of the technique.

9. What are the dynamics (time scale) being analyzed in the microtubule work?

10. A discussion of the limitations of the technique and the advantages of the technique in a separate section would be very helpful.

11. Several examples are provided of the techniques, although all of these demonstrate the technique it is a bit hard to understand how the technique works since the examples tend to be hard to follow. Refocusing the article more on the technique with fewer examples and provided better descriptions of the technique, including advantages and limitations, would better illustrate BARCODE.

12. What data is needed to input into BARCODE? Are there limited data types or formats that are needed?

13. Is there a minimum amount of data needed to provide significant results?

14. I understand the format of the article and why specifics of BARCODE are put in the SI, but I encourage the authors to move some of this to the main text. Since this is a technique development manuscript, the results are the technique and having this information in main text would make the manuscript more accessible. I would recommend moving this into the results of the main text and moving an example or two into the SI.

(Remarks on code availability)

Version 1:

Reviewer comments:

Reviewer #1

(Remarks to the Author)

The authors responded satisfactorily to my questions

(Remarks on code availability)

Reviewer #2

(Remarks to the Author)

I am quite happy with the revised manuscript. I especially like the discussion of photobleaching, and of the circular statistics (a term which I had not heard before, but which I just learned a fair bit about from the internet). I do have two minor concerns which the authors can address if they wish; these are optional.

1. I am still a little concerned about the Image Binarization branch. It seems like this is one of the few places in the BARCODE algorithm where there is user discretion. The Discussion section says "To ensure BARCODE is broadly accessible, we designed it to require minimal subjective inputs..." and this is true and a strength; nonetheless, results may vary based on those subjective inputs. I'm still unclear on exactly how sensitive the results could be between different users who are making good-faith efforts to analyze their data correctly, but who nonetheless have different judgement calls and/or levels of experience. I'm not even sure my own ability to threshold an image is consistent from day to day. (Although I agree that the 10% default in the BARCODE software is sensible and will ensure consistency for many users.) I would guess the authors are reluctant to add a caveat to the Discussion section or elsewhere in the main text, but I suggest adding some statement to the Supplemental Information that notes the results will depend on the choice of the offset. Or perhaps a recommendation in the software itself that suggests users stick with the default when possible.

2. I have a minor question about supplement Sec. 2.2. It states "A high skewness value indicates that the distribution has a long tail on one side, while a low skewness value suggests symmetry in the intensity distribution around the mean." This is slightly incorrect. $S = 0$ suggests symmetry. $S < 0$ is possible and would suggest asymmetry of the opposite sort of $S > 0$. I did a quick numeric check. If most pixels are value 1 but 1% are value 0, then $S = -9.7$. If instead 1% are value 2, then $S = +9.7$. And to confirm, if 1% are value 0 and another 1% are value 2, then $S = 0$.

(Remarks on code availability)

Reviewer(s)' Comments to Author:

Reviewer #1 (Remarks to the Author):

The main challenge addressed in the article, as I see it, is managing large-scale data—both in terms of enabling effective collaboration (a term I personally prefer over "democratization") and in extracting meaningful insights despite limited computational capacity due to data volume. This is indeed a critical problem, and addressing it can significantly advance scientific research.

We appreciate the reviewer's positive assessment of the potential impact of our work. Below, we respond to each question in detail in blue. We have also included below, and in a marked copy of the manuscript, highlights of the sections we revised in dark orange.

However, my primary concern lies in the selection of parameters. Specifically, I question whether using only 17 parameters is sufficient to capture the complexity of such a large dataset. One potential solution would be to start with a broader set of features and then apply dimensionality reduction techniques. If a larger set of features were used, alternative methods of visualization might also be necessary.

Response:

We agree that parameter selection is important, and appreciate the opportunity to provide additional details. Our team developed the 17-parameter barcode based on our extensive analysis of a wide range of active matter systems, including contractile protein networks and cell monolayers. We found these 17 metrics, derived from 3 parallel analysis branches, were able to capture key dynamic and structural information with manageable computational complexity, thereby balancing our desire to obtain useful insight into material properties while maintaining the ability to rapidly screen through large datasets. Moreover, we also provide users access to the full reduced data structures (RDS) which are the intermediate level datasets from which the 17 metric barcode is produced. For example, the RDS include the binarized images, not just the island size or connectivity metrics, as well as the full intensity distributions and flow fields, from which the statistical measures (means, standard deviations, etc.) that appear in the barcode are calculated and reported.

We added the following text on page 4:

We compute all 17 metrics (Table 1) using standardized reduced data structures (RDS) (Fig. 1D) –binarized video images (IB), pixel intensity distributions (ID), and velocity fields (OF) – which are automatically produced and archived to facilitate future downstream analysis and that contain substantially more information than the 17-parameter barcode itself. We

envison the barcodes will be used primarily for initial assessment and rapid down-selection, and the standardized outputs of the RDS will facilitate more time-intensive and hypothesis-driven downstream analyses. Based on our extensive analysis of active matter systems, we found that BARCODE's reduced set of 17 simple metrics captures key dynamic and structural information during rapid screens of large datasets.

These RDS are reduced in dimension with respect to the initial video data, but contain substantially more information than the 17 parameter barcode itself. We also enable users to control the level of downsampling, in both space and time, which is used in the calculation of the binarized images in the IB branch and velocity vector fields in the OF branch. We envision the barcode output will be used primarily for initial assessment and rapid downselection, and the standardized output of the RDS will facilitate more time-intensive and hypothesis-driven downstream analyses. We have emphasized these aspects for the reader in the extensively revised section on “Demonstrating BARCODE workflow and utility” that starts on page 5 of the revised manuscript. We refer the reviewer to the marked copy of the manuscript to review these changes in their entirety.

We understand that as BARCODE is applied to additional material systems, it is likely that additional metrics may be required, or some of the existing metrics may not prove as useful. To proactively address this issue, we constructed BARCODE in a modular fashion, allowing the expansion of existing branches or even the addition of new branches to enable the functionality and performance of the platform to grow over time. We believe this modularity is a key strength of the method, and we are happy to provide additional details to clarify this for the reader. We include this statement on p. 4 where BARCODE is first introduced:

However, the software is material-agnostic, modular and highly adaptable: branches and metrics can be easily added or removed without impacting other metrics, thus providing flexibility, while allowing for discovery of unexpected correlations or trends.

Finally, we agree that further dimensionality reduction through machine learning approaches may provide additional benefit, and this is an area of ongoing research in our groups. We have added a statement describing these future directions to the Discussion on page 19.

BARCODE is also highly adaptable: branches and metrics can be easily added or removed, providing flexibility to users and enabling the software to meet the future needs of the community. Planned expansions of BARCODE will include additional branches and metrics that quantify diffusive behavior and mechanical properties, including two-point correlation

functions, as well as adding user-identified improvements and user-generated extensions in the future. We are also currently investigating the use of machine learning for data classification and to enable data-driven predictions of material behavior.

Additionally, I would suggest outlining potential next steps in leveraging big data and machine learning techniques. One such step could be the automatic extraction of relevant features or parameters.

Response:

Thank you for the encouragement to add additional details about next steps for this work. As mentioned above, we are already working to apply machine learning approaches to dimension reduction and clustering of BARCODE outputs, which we hope to describe in a future publication soon. We have also started exploring options to extract information directly from the raw video files using data reduction and compression methods which may allow us to identify and extract embedded information automatically, although this work is still in the very early stages and we are not ready to comment on its potential for improvement at this stage.

In summary, I believe the article could benefit from better contextualization for readers who are less familiar with data science. It should also give more critical attention to the implications of representing a massive dataset with a relatively small number of parameters.

Response:

As outlined above, we have modified the discussion section to include more details of the implications of and opportunities provided by the data reduction approaches used here. We have added a new description of the advantages and disadvantages of our method in the Discussion section on pages 18-19. We also included an expanded description of how the data is inputted and reduced, including explicit references to our graphical user interface (GUI) and online tutorial, which provides additional information, on page 4:

As fully described in the Supplemental Information (SI, Section 1), BARCODE is a Python-based package that reads .tif and .nd2 video files and converts the data into arrays with dimensions (T,m,n,c) , where T is the number of frames, m and n are the number of pixels along the horizontal and vertical axis of each frame, and c is the number of channels (for, e.g., confocal videos with multiple components of a material labeled with distinct fluorophores and recorded in separate detectors). The software has a user-friendly graphical

user interface (GUI) and several adjustable parameters that the user can set to tailor and optimize the operations of each branch for their system. We also provide a detailed online tutorial⁴⁸ to guide users in choosing parameters that best suit their data. BARCODE outputs include three RDS files, one for each branch, for every video, saved as .csv files; and a .csv file and colorized .svg with the 17 BARCODE metrics for the entire dataset (e.g., Figs. 1E, 2E).

Reviewer #2 (Remarks to the Author):

This manuscript presents a reasonable straightforward method called BARCODE to characterize several interesting properties of microscopy videos, which can then be used to compare videos from different experiments. The method is flexible as regards to the number of images and their exact dimensions, which is great. Several example data sets are used as demonstrations that the method is powerful. I think this is publishable after the authors clarify some points.

Response: We appreciate the positive feedback and helpful comments that follow. Below, we respond to each question in detail in blue. We have also included below and in a marked copy of the manuscript highlights of the sections we revised based on reviewer input in dark orange.

1. I'm most concerned about the Image Binarization branch. There is one user-defined threshold ('offset' defined in supplemental [sic] section 1). It seems that the results will be quite sensitive to this choice; that for example, the connectivity depends on this choice. Of course, BARCODE has far fewer parameters than many competing methods, so this one user-defined parameter is not a fatal flaw. Nonetheless, the authors should comment on how users might pick this threshold, and how sensitive the results can be on this choice. For example, which of the results presented in Figs. 2-5 would change significantly based on the choice of the threshold? The author list is long, so if different authors independently looked at the images and chose thresholds, how much variability occurs? Again: this is not a huge problem, but I suspect that different data will have different sensitivity to these choices, and different people might make different choices, so the ability to compare across data sets and between different research groups would potentially be diminished.

Response:

We agree that the user-defined threshold is an important parameter and are happy to provide additional details. In our initial algorithm designs, we performed a sensitivity analysis to define the default threshold parameter, and determined that using a threshold of 10% above the mean intensity (we term this 10% value the 'offset'), determined on a per-frame basis, was a good default value that generally worked well for the datasets presented in the manuscript.

We chose to use the frame-specific mean intensity to minimize any possible effects of photobleaching. However, we agree that each dataset is unique and may require some user-based optimization. To enable this optimization, we allow users to set their own thresholds, and since BARCODE is designed to run quickly, it is possible for users to perform their own rapid sensitivity analysis, to inform their own choices of offset values. To facilitate this process, we have modified the graphical user interface to enable users to test different threshold values and preview the resulting binarized images. This feature allows users to rapidly assess the threshold sensitivity and choose an offset value that captures the desired structures and dynamics within their own data. We have included additional details describing how users can perform these steps to optimize the value of threshold for their own data in the revised SI and have updated the online tutorial, which is available at the Github site (link: <https://github.com/softmatterdb/barcode>). The following description was added to page 5:

The threshold pixel intensity for binarization is computed from a user-specified offset that provides a threshold percentage %I above the mean intensity of each image (Fig. 2B, top). This process assigns a unique threshold intensity value for each frame, set by its mean value, which corrects for reduced mean intensities of images over time due to photobleaching and other spurious fluctuations in intensity.

2. For the Intensity Distribution branch, how does photobleaching affect the results? In the paragraph on page 4 introducing the ID branch, it interprets the decrease of the mean intensity in Fig. 2C as due to the formation of aggregates. In some of my data sets, it would be due to photobleaching, which is not the same science as aggregation. How much is this a problem?

Response:

We appreciate the reviewer's thoughtful question, and agree that managing the treatment of photobleaching, which is often observed in the visualization of active biomaterials, is important to achieving reliable and interpretable results. We are happy to provide additional details.

We used several methods to manage the potential for photobleaching in designing BARCODE. As described above, within the Image Binarization branch, we detect the mean intensity in each frame of the movie, and use the frame-specific mean intensity value to set a frame-specific default threshold value. This ensures that we can reliably detect structures even if the mean (background) intensity changes over time. Separately, we flag videos that we deem too dim for reliable analysis. "Dim" videos are defined as those where the first and last frames have minimum pixel intensity value greater than or equal to $2/e$ times the mean pixel intensity value for that frame, the video will have a Flag value of 1. Users have the ability to either include

these files (which are flagged in the output CSV files) or to ignore them in the analysis. Saturated videos (defined as the case where the mode pixel intensity value is equal to the maximum pixel intensity value for all frames), are also separately flagged.

In the Intensity Distribution (ID) branch, photobleaching is accounted for in the determination of skewness and kurtosis, which are measures of distribution symmetry rather than a simple assessment of the distribution mean. In general, one would expect that if photobleaching is the dominant effect that the mean of the intensity distribution would decrease with time, shifting the overall distribution left towards zero. However, the shape of the distribution would likely not change substantially, since the underlying distribution of structures is not changing. By contrast, material restructuring due to aggregation, phase separation or bundling should change the distribution of mass, and therefore intensity, and should lead to changes in the distribution shape, which are detected via the kurtosis and skewness outputs of BARCODE.

In the case of severe photobleaching, one might expect that eventually some fraction of low intensity pixels would become so dark that their intensity would not be detectable over the static pixel noise which arises from e.g. stray light, dark camera noise, etc. In this limit, we would expect the shape of intensity distribution, and therefore the kurtosis or skewness, to likely change due to our inability to detect the full distribution (i.e. due to 'missing events'). A scenario where a substantial fraction of pixels fall within this undetectable limit would therefore be problematic and may lead to unreliable results. This is exactly why we flag frames and videos with low intensity (described above). Additionally, we provide two measures of skewness, based on both the median and mode, which have different sensitivities to a large number of extremal values in the distribution. We have rewritten the description of the IB and ID branch design to include more details about photobleaching in the revised manuscript to clarify this, including this description on page 7.

In general, if photobleaching were significant, one would expect the mean of the intensity distribution to decrease with time, shifting the overall distribution left towards zero. However, the shape of the distribution would likely not change substantially, since the underlying structures are not changing. By contrast, material restructuring due to aggregation, phase separation or bundling should change the distribution of mass, and should lead to changes in the distribution shape, which are detected via the kurtosis and skewness outputs of BARCODE. In the case of severe photobleaching, one might expect that eventually some fraction of low intensity pixels would become so dark that their intensity would not be detectable over the static pixel noise which arises from e.g., stray light, dark camera noise, etc. In this limit, we would expect the shape of the intensity distribution, and therefore the kurtosis or skewness, to likely change due to the inability to detect the full

distribution (i.e., due to 'missing events'). To avoid this potential artifact, we flag videos with unusually low intensity and remove these from the analysis; saturated videos are also flagged (see SI Section 1).

With regard to the data shown in Fig 2C, we agree that our original description did not accurately capture the distribution features. We appreciate the reviewer's careful reading. We have corrected this, and substantially rewritten the associated section in the revised manuscript to improve clarity.

3. I'm also wondering how much uneven illumination would affect both IB and ID questions. Some days my fluorescent arc lamp is better aligned than others.

Response:

We encourage users to collect the highest quality data possible and to avoid uneven illumination or other artifacts, as we agree that they could lead to unintended consequences in analysis. We have added the following statement to page 5 to clarify this for the reader.

We assume in this analysis that there are minimal spatial variations in background intensity across the field of view.

In cases where uneven illumination cannot be avoided, one may need a matrix of threshold values in the binarization branch and a more complex statistical approach to assess changes in the intensity distribution, in order to compensate for poor image quality. While technically possible, this would require significant changes in the calculation of the various BARCODE outputs and/or normalization with a control image(s), which would introduce significantly more computational complexity, which would undermine the HTP aspects of BARCODE for which it was designed.

4. For the flow direction theta, it took me a while to understand why this is useful. The point is that if one has a two channel image, if both channels indicate the same flow direction, then the two components being visualized are co-moving. That's great. This should be explained, because if I'm just comparing across multiple experiments, the particular angle theta seems irrelevant. One could change it just by rotating the sample or camera, presumably.

Response:

We thank the reviewer for raising this point, and appreciate the opportunity to provide additional details. We agree that flow direction is a parameter that may require some knowledge of the material to interpret, and that flow direction may not be a relevant parameter for all materials. We agree that the identification of co-flowing composite materials is one important outcome. The flow direction metrics also allow us to distinguish directed motion from isotropic motion. As described in more detail in our answer to Question #5 that follows, for directed motion, the flow direction θ indicates the average direction of flow, and the directional spread σ_θ should be approximately zero, since all vectors are pointing in the same direction. By contrast, for isotropic motion the flow direction θ should average to zero and the directional spread σ_θ should be >0 . We have included additional information in the manuscript in the description of the OF branch outputs on page 8 to address this point, and have updated the SI and tutorial accordingly.

In terms of the physical meaning of any particular angle, this is very experiment-specific, and the flow angle can provide physically meaningful information. We have found that many active networks are influenced by the chamber geometry and presence of an aligned side wall of the chamber. For our protein composite work, we often use long channeled chambers, with a large aspect ratio of channel length to width or height, and align this long axis to the microscope stage (and camera) in order to set a physically meaningful angle (in our experiments, this long axis aligns to $\theta = \pi/2$) It is also possible to use textured or architected substrates to control the directions of active matter flows. These examples are beyond the scope of what we analyzed and presented in this manuscript, but we wanted BARCODE to provide capabilities for this type of analysis. We have expanded the discussion of the data presented in the caption of Figure 3f to the revised manuscript on page 11 to describe this.

5. But more importantly for theta, how does the software handle the definition of theta as between 0 and 2 pi? That is, suppose the motion is close to theta = 0. Fluctuations of the direction might mean that sometimes the direction is plus or minus a few degrees; but negative angles will be close to 2pi. The average of 0 and 2pi is pi, the opposite direction of the actual motion. Moreover, the standard deviation (Eqn. 27) will be much larger than it should be. How is this handled for both computing mean theta and computing the standard deviation?

Response:

We appreciate this question. While this was not a scenario we encountered in testing BARCODE, we agree this could be an issue for the motion at or near an extremal value of angle, which range from $-\pi$ to π . To address this possibility, we have redefined the calculation of

average direction to use circular statistics, which avoid the issue of periodicity at the extremal values of angle. In calculating the mean, we changed the order of operations to take the mean of the x and y components of velocity first, before taking the arctangent. We also implemented use of the two-argument $\arctan2$ function which uses the signs of both the average x and y components of the velocity vector to determine the correct quadrant. We have validated this implementation using simulated flow fields that are aligned to the extremal angle range. Similarly, we now use the circular standard deviation to calculate the directional spread. We have updated the text and SI accordingly.

6. Related to this, Fig. 3F(inset) caption says "fast flows are directed along plus/minus pi." This seems like it must be an artifact -- it depends on the camera orientation -- why would plus or minus pi be special? Although looking closely at the inset, I'm not seeing how "plus/minus pi" is deduced. pi is 3.14 and the horizontal axis of the inset ranges from -2 to +2, so does not include plus or minus pi at all. I am really confused.

Response:

We appreciate the reviewer's careful reading. As we noted above, active networks are often influenced by the chamber geometry and presence of an aligned side wall of the chamber. For our protein composite work, we often use long-channeled chambers, with a large aspect ratio of channel length to width or height, and align this long axis to the microstage (and camera) in order to set a physically meaningful angle. In our experiments, this long axis aligns to $\theta = \pi/2$. The plot in Figure 3F is bounded to reflect this preference for fast flows to align to $\theta = \pi/2 \sim 1.5$ radians; there were no detected values greater than 2 radians or less than -2 radians in this dataset. We have corrected the caption and added the following explanation of the boundary-guided flow alignment $\theta = \pi/2$.

Correlation between σ_θ and flow direction θ shows fast flows are directed along $\pm \frac{\pi}{2}$ due to presence of the chamber side wall that is aligned to that direction, while slow and multi-mode videos have no preferred direction.

7. In Fig. 3B caption it talks about "temporal colormaps." I do not understand what these are, please define.

Response:

We thank the reviewer for raising this point and we agree that the caption could be improved. Figure 3B shows an array of small panels, representing the three dynamic classes: slow, fast and multi-mode. For each of these three classes, we show three panels, a full-frame 2-channel

image (left) showing the actin (green) and microtubule (magenta) structures at the start of the time-series ($t = 0$). To the right of this panel, we show temporal colormaps in two half-frames of the microtubules only (top right) and actin only (bottom right). For each temporal colormap, images of the network obtained at different times are generated and overlaid, with the image color varying from blue at $t = 0$ to red at $t = 6$ minutes, as shown in the color legend bar at the top of each temporal colormap pair. With this color-mapping approach, network motions can be clearly identified from the single image, as the colors change from blue to red. Large-scale streaks or lines within this temporal color map indicate regions of large-scale directed displacement over time. Color maps showing many colors but no directed streaks indicate more random, isotropic motion. Color maps with little to no color indicate that the sample lacks dynamic motion over the timescales observed. We updated the caption of Figure 3B to clarify.

(B) Multi-channel images of actin (green) and microtubules (magenta), as well as single-channel color-coded temporal projections, in which a 6-min video is collapsed to a single image with features from each frame distinctly colorized according to the time-color scale shown.

8. The supplemental materials are nice, and generally tell exactly how to compute the various quantities. The exception is the kurtosis: please give a formula for how the kurtosis is calculated. The reason I'm asking is that I normally think of skewness as calculated based on moments of the distribution, but BARCODE is instead using "median skewness" and "mode skewness" which are not based on moments. That's fine, but it makes me wonder if the kurtosis is calculated based on moments. That is, is the kurtosis the fourth standardized moment defined on Wikipedia (<https://en.wikipedia.org/wiki/Kurtosis>)? And why not use the third standardized moment for defining skewness (<https://en.wikipedia.org/wiki/Skewness>)? I don't feel strongly about any of this, other than to just simply include an explicit formula for kurtosis in the supplemental materials.

Response:

We apologize for the oversight. We have modified the text on page 6 of the manuscript as copied below, and have updated the SI accordingly.

Kurtosis, computed as the fourth central moment of the distribution normalized by the squared variance, $K = \frac{\mu_4}{\sigma^4} - 3$, reports the extent to which pixel values are closer to ($K < 0$) or further from ($K > 0$) the mean than expected for a normal distribution (see SI, Section 1). Positive kurtosis values are indicative of de-mixing, coarsening and/or clustering while negative values indicate uniform and/or space-filling materials.

Reviewer #3 (Remarks to the Author):

This article describes a new analysis technique called BARCODE. This technique takes large data sets of videos and is able to process it quickly into many different variables that are then tested to see whether they are correlated. The text describes several examples from already analyzed data sets to show the utility of the technique and also shows how BARCODE can be used on new data to find new correlations between variables. This is a really interesting technique that could be impactful especially with the amount of data that can be collected steadily increasing. The manuscript is written in a way that makes it somewhat hard to understand the technique. As mentioned below, I think using less examples and treating the development of the technique as part of the results and providing a more thorough description in the main text would greatly improve the manuscript. I read the manuscript first and then the SI and it was not until I read the SI that I really started to understand how BARCODE worked. Not all details in the SI need to be added to the main text, but some descriptions of the technique do need to be moved to the main text. This can be done by moving an example or two to the SI. Due to this and other comments provided below, major revisions are required prior to publication in Nature Communications.

Response:

We appreciate the reviewer's constructive feedback, and are happy to provide additional context. We have extensively modified the manuscript and SI to include more details and descriptions of the techniques we use and the different parameters we compute, and to address each specific point raised by the reviewer. Below, we respond to each question in detail in blue. We have also included below, and in a marked copy of the manuscript, highlights of the sections we revised in dark orange. We also now include much more detail about the physical meaning of the BARCODE parameters and important user considerations, and have streamlined the discussion of the individual examples to address the reviewer's suggestions.

Comments, questions and concerns:

1. The three channels chosen for independent analysis seem to be limiting to the technique. What is the rationale for choosing these three channels and what kind of information is excluded by choosing these?

Response:

We appreciate the opportunity to provide more details. In selecting these three branches, we focused on how we could leverage established image analysis methods to rapidly access the most important information of material structure and dynamics and enable comparison and

downselection for additional screening and analysis of common active materials. We also aimed to improve a user's ability to more easily share reduced datasets among collaborators and the community, in the form of the 17-metric barcodes and reduced data structures (RDS) that encoded easily understandable information. We did not seek to fully characterize the active materials with BARCODE alone, rather we aimed to provide reliable and actionable information to users to enable decision making with regard to future experiments or analyses, and provide preliminary insights. By using established image analysis methods to generate the RDS, we aim to empower users, including non-experts, to more readily analyze their results and compare their outputs to existing literature values.

To clarify this for the reader, we added this sentence to page 4, as well as new references #41-47, as listed in the revised manuscript:

We designed each branch to leverage established image processing tools—image binarization (IB), pixel intensity distribution (ID) analysis, and optical flow (OF)⁴⁰—for which there is an extensive body of literature describing approaches, implementation, and best practices⁴¹⁻⁴⁷.

Additionally, we added the following, also on page 4:

We compute all 17 metrics (Table 1) using standardized reduced data structures (RDS) (Fig. 1D) —binarized video images (IB), pixel intensity distributions (ID), and velocity fields (OF) — which are automatically produced and archived to facilitate future downstream analysis and that contain substantially more information than the 17-parameter barcode itself. We envision the barcodes will be used primarily for initial assessment and rapid down-selection, and the standardized outputs of the RDS will facilitate more time-intensive and hypothesis-driven downstream analyses. Based on our extensive analysis of active matter systems, we found that BARCODE's reduced set of 17 simple metrics captures key dynamic and structural information during rapid screens of large datasets. However, the software is material-agnostic, modular and highly adaptable: branches and metrics can be easily added or removed without impacting other metrics, thus providing flexibility, while allowing for discovery of unexpected correlations or trends.

In detail, the 17-parameter barcode that is produced from the 3 branches was developed based on our extensive analysis of a wide range of active matter systems, including contractile protein networks and cell monolayers. We found this set of metrics captured key dynamic and structural information with manageable computational complexity, thereby balancing our desire to obtain useful insight into material properties while maintaining the ability to rapidly

screen through large datasets. Moreover, the standardized RDS outputs provide additional and richer information, enabling key insights beyond those captured by the 17-metric barcode. The IB branch focuses on structural changes within the material, through analysis of connectivity and the formation of discrete islands (or voids). The ID branch provides information about changes in intensity distribution, which is representative of the redistribution of mass within the material, potentially as a way to assess material compaction or aggregation dynamics. The OF branch assesses the speeds and directions of motion of the imaged material. We have emphasized these aspects for the reader in the extensively revised section on “Demonstrating BARCODE workflow and utility” that starts on page 5 of the revised manuscript. We refer to the reviewer to the marked copy of the manuscript to review these changes, which are marked in dark orange in the text.

We understand that as BARCODE is applied to the additional material systems, it is likely that additional metrics may be required, and/or some of the existing metrics may not prove as useful. To proactively address this issue, we constructed BARCODE in a modular fashion, allowing the expansion of existing branches or even the addition of new branches to enable the functionality and performance of the platform to grow over time. We believe this modularity is a key strength of the method, and have added more details to the manuscript to make this clearer for the reader – in particular, see p. 4 where BARCODE is first introduced and as copied above.

It is difficult to answer the question of what is excluded, as there are many potential paths for analysis that we did not fully explore in this initial work, and enumerating everything *not* done is not straightforward. We can comment that we are currently testing an additional branch to improve the analysis of diffusive dynamics and anisotropies, leveraging aspects of differential dynamic microscopy and particle tracking. We also plan to expand the scope of the three existing branches to enable two-point correlations between two spatially separated zones, which would begin to allow us to assess how strains, alignment, and even stresses might be correlated over space and time. This would provide new insight into material dynamics and may enable integration of BARCODE outputs into computational or analytical modeling. We are also developing deeper analyses, albeit at lower throughput, of the RDS. All of these expansion efforts are in the early phases, but we have added a few comments to the discussion on pages 18-19 to better describe these future directions for the reader and to emphasize that BARCODE is an open-source platform that we anticipate will expand and evolve over time, based on inputs and innovation from our collaboration and the community, including the following:

BARCODE is also highly adaptable: branches and metrics can be easily added or removed, providing flexibility to users and enabling the software to meet the future needs of the community. Planned expansions of BARCODE will include additional branches and metrics that quantify diffusive behavior and mechanical properties, including two-point correlation functions, as well as adding user-identified improvements and user-generated extensions in the future. We are also currently investigating the use of machine learning for data classification and to enable data-driven predictions of material behavior.

2. A colorbar should be provided for Fig. 1E, even if it is just qualitative.

Response:

We originally included a color bar showing the color distribution from the maximum and minimum values of each metric in Figure 2, but agree that including one in Figure 1 would be helpful. We have updated the manuscript to include a similar color bar at the bottom left hand side of Figure 1E. We also included the numeric range of minimum and maximum values for each metric under the barcode output just next to the name of the metric displayed. We have also updated the caption to make this clearer to the reader.

3. There seems to be some redundancy in the first paragraph of the Results, the 3 image processing tools and the reduced data structure, I think are the same but are both listed a sentence apart, are they the same and is this redundant?

Response:

We thank the reviewer for raising this point and we agree that there is some redundancy in the paragraph. We have modified this description in the revised text to improve clarity.

4. How does reducing the data change the accuracy of the results?

Response:

We agree that the significant dimension reduction through BARCODE, especially in the 17-parameter barcode, inevitably leads to some information loss. However, we have taken care to select metrics that we have found can provide actionable physical insight, even in this limit. Moreover, we have designed BARCODE for use in HTP screening to enable inline experimental

decision making and facilitate down-selection for deeper analysis, therefore we have designed for speed and computational simplicity rather than fidelity, accuracy and precision. That said, we have found good agreement between the BARCODE outputs and other published results. In some cases, we have been able to perform quantitative comparisons, for instance of the average speeds of restructuring of active cytoskeletal composites, as shown in Figure 3E. Here, we found excellent agreement between the mean speeds v computed from BARCODE as compared to the previously reported values computed using Dynamic Differential Microscopy (DDM). In analyzing the full published dataset, we also found that the directional spread σ_θ and flow direction θ outputted by BARCODE not only reproduced the published results on a small subset of the data, which were determined using particle image velocimetry (PIV), but also demonstrated the universality of the results for all data in each dynamic class. This result was not previously reported due to the labor-intensive data processing that prevented PIV analysis from being able to be performed on the entire dataset in a tractable time frame (Fig. 3F).

We have additionally observed qualitative agreement with previously published results, for example, in assessing connectivity and the (inverse) correlations between percolation and flow dynamics. We have added the following on page 12 to further clarify this for the reader:

To corroborate this physical picture, we compare the connectivity and maximum void area for the different classes (Fig 3J), finding that nearly all slow and multi-mode networks remain fully connected ($C \approx 1$) and most void areas are below 50%. Conversely, BARCODE reports a range of fractional ($C < 1$) connectivity values for the fast class networks as well as for microtubules, which also have larger void areas ($V > 50\%$) and smaller initial island areas compared to actin.

Further, to enable users to test the effects of downsampling of raw images to increase throughput, we have added additional details of how downsampling can be executed to balance accuracy and speed on page 5.

To increase processing speed and reduce the data size, the resulting stack of binarized images, which is a saved RDS, can be downsampled in time, by choosing to analyze every k^{th} frame, and spatially, by averaging together $p \times p$ windows of pixels to result in a stack of T/k images of $m/p \times n/p$ pixels. We have found that $\%I = 10$, $k = 10$ and $p = 2-8$ to provide sufficient resolution and accuracy while maintaining rapid processing times (SI Table S1).

We have added additional details regarding these design choices and comparisons, in the “Demonstrating BARCODE workflow and utility” section that starts on page 5 and that we have extensively revised. We anticipate that different material systems will demonstrate different

sensitivities to dimension reduction, so we hope these additional details will empower users to test for sensitivity to sampling parameters directly in their own data.

5. It seems that time-dependent data is lost in the data reduction, is that correct? If this is correct, how does this limit the method?

Response:

No, this is not correct, and we appreciate the opportunity to clarify. In fact, many of the barcode parameters assess time dependence by reporting the change in the average value of a metric of interest computed for the last 5% of frames in a time series versus the first 5%. These parameters include: Maximum Island Area Change, Maximum Void Area Change, Kurtosis Change, Mode Skewness Change, Median Skewness Change, and Speed Change. Other metrics, such as the Connectivity and Flow Direction Spread are also sensitive to the entire time trajectory of the video. Additionally, in all branches we provide the RDS as a function of time for the entire time trajectory of each video, not just the first 5% and last 5%. We allow users to downsample the video in space and time to improve throughput; for the systems we describe within this work, we find using every 10th frame generally provides a good balance of information and speed.

We view these time-dependent metrics and features of BARCODE to be essential, since capturing time-dependent changes in material structure and dynamics is very important for active systems. We have added additional details emphasizing these features for the reader. We now include this statement on page 2:

To produce each barcode, the software also calculates rich reduced data structures (RDS) that enable more detailed understanding of the time-evolving material structures and mechanics, and archives the RDS to enable subsequent hypothesis-driven research.

And we added this description to the Discussion on page 19:

To increase processing speed and ensure the metrics are applicable to a broad range of materials, we chose to reduce spatial and temporal resolution (downsampling frames and pixels by k and p) in calculating the RDS, and to focus on extremal quantities and changes thereof. In general, the metrics that report changes in quantities assess property differences between the beginning and end of each video, so transient changes may not be captured in the lowest dimension barcode matrix. However, the full time-

dependent information is saved in the RDS and users can choose to perform advanced analysis on RDS to assess time-dependencies and discern more subtle changes in material structure and dynamics.

We have also included more information in the revised section on Demonstrating BARCODE workflow and utility section that starts on page 5 to make this clearer for the reader. We have indicated in the SI what inputs are user-defined in the current implementation of BARCODE. We will also maintain a current list of user-defined quantities in the tutorial, which will capture any changes or additions that are made in the future.

6. What are the arrows in Figure 3E?

Response:

The arrows indicate the measured outputs for the networks shown in panel 3B. We have added the following statement to the caption and labeled the arrows to clarify this for the reader.

The arrows indicate the data points corresponding to the conditions shown in panel B.

7. If the changes in the images are not in a single direction, would this method work?

Yes. The flow direction metrics allow us to distinguish directed motion from isotropic motion. For directed motion, the flow direction θ indicates the average direction of flow, and the directional spread σ_θ should be approximately zero, since all vectors are pointing in the same direction. By contrast, for isotropic motion the flow direction θ should average to zero and the directional spread σ_θ should be > 0 and should increase with increasing disorder. Other flow types would result in metric values intermediate between these extremal cases. In cases where the barcode is difficult to interpret directly, the OF RDS outputs will allow the user to visualize changes in the flow field on a per-frame basis to improve understanding. We have included additional information in the manuscript in the description of the OF branch outputs on page 9 to address this point, and have updated the SI and tutorial accordingly.

8. It is hard to follow the examples provided since they are described in terms of the variables and not well connected to the physical results in the movies. The actin-myosin example did a better job of connecting the variables to the physical changes in the videos but some discussion

was still very hard to follow. It would help if the variables being described are related to what they physically represent in the videos, that would aid in the readability of the manuscript and better illustrate the power of the technique.

Response:

We have rewritten the results section of the manuscript to add context and help the reader interpret the BARCODE outputs in light of the physical parameters of the various materials. Newly added text is marked in dark orange in the revised manuscript.

9. What are the dynamics (time scale) being analyzed in the microtubule work?

Response:

The dynamics in the microtubule networks arise from the kinesin motors crosslinking and pulling microtubules relative to another. As we describe in the SI Methods section, each time-series analyzed to produce the results shown in Fig 4H-M was collected at 1.33 frames per second for a duration of 300 s (4000 frames). From BARCODE, we measure average speeds of the microtubules in the range $\sim 10^{-1}$ to ~ 10 $\mu\text{m/s}$.

10. A discussion of the limitations of the technique and the advantages of the technique in a separate section would be very helpful.

Response:

Thank you for the encouragement to add additional details about the advantages and limitations of the technique. We have rewritten the discussion to include these details, see page 18-19.

11. Several examples are provided of the techniques, although all of these demonstrate the technique it is a bit hard to understand how the technique works since the examples tend to be hard to follow. Refocusing the article more on the technique with fewer examples and provided better descriptions of the technique, including advantages and limitations, would better illustrate BARCODE.

Response:

We appreciate this constructive criticism and have substantially rewritten the manuscript to address these concerns. We have added a new description of the advantages and disadvantages on pages 18-19, and have provided more background and physical interpretation of the data obtained from the various material sets. We appreciate the reviewer's suggestion to remove examples from the study, but we think it is important to include them all to demonstrate the full power of BARCODE to analyze a wide area of protein- and cell- based active materials. However, to address the reviewer's suggestion we have substantially simplified and streamlined our discussion of each example, including simplifying the figures; and we have revised the text substantially to provide more description of the technique and contextualization of the results, as described in detail above. New text added in the revision has been marked in dark orange in the revised manuscript.

12. What data is needed to input into BARCODE? Are there limited data types or formats that are needed?

Response:

In the current iteration of BARCODE, we can accept tif stacks and .nd2 files as inputs; however, it is straightforward to incorporate other files types in our I/O structures, and we are currently beta testing functions to read and analyze .avi and .czi files as well. We intend to describe all improvements to the code and all versions on our Github site, but have not yet released these for general use and prefer not to describe these changes in detail in this manuscript.

To better clarify this for the reader, we have added this description on page 4:

As fully described in the Supplemental Information (SI, Section 1), BARCODE is a Python-based package that reads .tif and .nd2 video files and converts the data into arrays with dimensions (T, m, n, c) , where T is the number of frames, m and n are the number of pixels along the horizontal and vertical axis of each frame, and c is the number of channels (for, e.g., confocal videos with multiple components of a material labeled with distinct fluorophores and recorded in separate detectors). The software has a user-friendly graphical user interface (GUI) and several adjustable parameters that the user can set to tailor and optimize the operations of each branch for their system. We also provide a detailed online tutorial⁴⁸ to guide users in choosing parameters that best suit their data.

13. Is there a minimum amount of data needed to provide significant results?

Response:

In terms of video size, videos of at least 5 frames are required, and the maximum file sizes are limited by the computational power of the analysis hardware and the time-constraints of throughput of the analysis. In terms of time-dependent quantities - the step size used to calculate the RDS for each of the three branches can be adjusted based on number of frames needed to give adequate sampling of the material dynamics. Spatial downsampling is also available. In general, inclusion of longer videos and more videos will enable stronger statistical confidence in the results, as expected.

To answer this question in detail requires several assumptions about data quality and material dynamics, which are difficult to justify without a specific material system in mind. For the materials we analyze and present, we find that videos of 256×256 size and 250 frames are generally sufficient to extract useful information. These details were included in Table S1 in the SI.

14. I understand the format of the article and why specifics of BARCODE are put in the SI, but I encourage the authors to move some of this to the main text. Since this is a technique development manuscript, the results are the technique and having this information in main text would make the manuscript more accessible. I would recommend moving this into the results of the main text and moving an example or two into the SI.

Response:

We appreciate the suggestion and agree. We have substantially rewritten the manuscript and in particular, the section 'Demonstrating BARCODE workflow and utility' to address this. New text added in the revision has been marked in dark orange in the revised manuscript.

Response to Reviewers

REVIEWER #1

The authors responded satisfactorily to my questions.

We thank the reviewer for their input and time.

REVIEWER #2

I am quite happy with the revised manuscript. I especially like the discussion of photobleaching, and of the circular statistics (a term which I had not heard before, but which I just learned a fair bit about from the internet). I do have two minor concerns which the authors can address if they wish; these are optional.

We thank the reviewer for their positive responses, input and time. We have addressed the two minor concerns below.

1. I am still a little concerned about the Image Binarization branch. It seems like this is one of the few places in the BARCODE algorithm where there is user discretion. The Discussion section says "To ensure BARCODE is broadly accessible, we designed it to require minimal subjective inputs..." and this is true and a strength; nonetheless, results may vary based on those subjective inputs. I'm still unclear on exactly how sensitive the results could be between different users who are making good-faith efforts to analyze their data correctly, but who nonetheless have different judgement calls and/or levels of experience. I'm not even sure my own ability to threshold an image is consistent from day to day. (Although I agree that the 10% default in the BARCODE software is sensible and will ensure consistency for many users.) I would guess the authors are reluctant to add a caveat to the Discussion section or elsewhere in the main text, but I suggest adding some statement to the Supplemental Information that notes the results will depend on the choice of the offset. Or perhaps a recommendation in the software itself that suggests users stick with the default when possible.

We appreciate the reviewer's perspective, and have made the following changes:

The description of the binarization threshold (offset) has been updated on our Github page (<https://github.com/softmatterdb/barcode/blob/main/README.md>; doi: 10.5281/zenodo.17585070) as follows:

Controls the threshold percentage of the mean which binarizes the image; offset parameter determines the binarization threshold for a given frame as $(1 + offset)\underline{B}(i)$, where $\underline{B}(i)$ represents the mean pixel intensity for frame i . Users are cautioned that changes to the offset value can affect the outputs of the Image Binarization branch.

The tutorial (available here: <https://www.livingbam.org/barcode-tutorial>) has been updated to read:

*The Binarization Threshold is an important parameter that determines what pixels within the image represent material, and should be set to 1, and which are background or noise, and should be set to zero. The Binarization Threshold is calculated for each frame as $((1 + \text{Offset}) * \text{Mean Pixel Intensity})$. The offset is user-defined. By default, BARCODE uses an offset of 0.1. Users should take care when adjusting this value, as it can affect the results of the Binarization Branch. The effect of the threshold offset selection is demonstrated below by transforming the raw image into the following binary datasets based on various threshold values. As the threshold decreases more noise can be seen while as it increases more features are lost. The best offset value for the following examples would be 0.1 or 0.25.*

2. I have a minor question about supplement Sec. 2.2. It states "A high skewness value indicates that the distribution has a long tail on one side, while a low skewness value suggests symmetry in the intensity distribution around the mean." This is slightly incorrect. $S = 0$ suggests symmetry. $S < 0$ is possible and would suggest asymmetry of the opposite sort of $S > 0$. I did a quick numeric check. If most pixels are value 1 but 1% are value 0, then $S = -9.7$. If instead 1% are value 2, then $S = +9.7$. And to confirm, if 1% are value 0 and another 1% are value 2, then $S = 0$.

We have modified the text as follows:

A high positive (or negative) skewness value indicates that the distribution has a long tail of high (or low) intensities relative to the mean, whereas a skewness value near zero indicates that the intensity distribution is more symmetric around its mean.